# Formation of synthetic RNA protein granules using engineered phage-coat-protein -RNA complexes

Naor Granik [1], Noa Katz[2], Or Willinger[2], Sarah Goldberg[2] & Roee Amit [2,3] ✉

Liquid-solid transition, also known as gelation, is a specific form of phase separation in which molecules cross-link to form a highly interconnected compartment with solid – like dynamical properties. Here, we utilize RNA hairpin coat-protein binding sites to form synthetic RNA based gel-like granules via liquid-solid phase transition. We show both in-vitro and in-vivo that hairpin containing synthetic long non-coding RNA (slncRNA) molecules granulate into bright localized puncta. We further demonstrate that upon introduction of the coat-proteins, less-condensed gel-like granules form with the RNA creating an outer shell with the proteins mostly present inside the granule. Moreover, by tracking puncta fluorescence signals over time, we detected addition or shedding events of slncRNA-CP nucleoprotein complexes. Consequently, our granules constitute a genetically encoded storage compartment for protein and RNA with a programmable controlled release profile that is determined by the number of hairpins encoded into the RNA. Our findings have important implications for the potential regulatory role of naturally occurring granules and for the broader biotechnology field.

Phase separation, the process by which a homogeneous solution separates into multiple distinct phases, has been connected to a wide range of natural cellular processes in virtually all forms of life[1–5]. In cells, phase separation results in the formation of membrane-less compartments containing a high-concentration mix of biomolecules (e.g., proteins, RNA, etc.), which are surrounded by a low-concentration solution. Generally, phase separations are classified by the different material states which can lead to multiple types of transitions (e.g., liquid-solid, gas-liquid, etc.). The forms commonly reported in cellular biology are broadly liquid-liquid and liquid-solid (e.g., gelation), however determining the exact mechanisms for phase separation in a living cellular environment is often challenging[5].

Liquid-liquid phase transitions can be distinguished from liquid-solid by the dynamical properties of the resulting condensates. Liquid-based condensates show rapid internal rearrangement of molecules, fusions between different condensates upon contact, and dependency on the concentration of the molecules in the condensed phase[6,7]. On the other hand, liquid-solid based condensates show none of the above qualities and are mainly dependent on the number of 'cross-linkers', which are points of contact between the molecules, rather than on the concentration of the molecules themselves[8–10].

Recently, Jain & Vale[11] reported on the formation of RNA granules both in vivo and in vitro, from highly repetitive RNA sequences associated with repeat expansion diseases. These RNA sequences, comprised of dozens of triplet-repeats of CAG or CUG nucleobases, form intramolecular hairpin structures[12], which facilitate multivalent intermolecular interactions. The RNA granules presented features associated with liquid-solid phase transition systems: a lack of internal mobility, virtually no fusion events, and dependence on the number of repeats in the RNA sequence (i.e., cross linkers) rather than the concentration of the RNA. These characteristics helped establish the granules as physical solids.

[1]Department of Applied Mathematics, Technion—Israel Institute of Technology, Haifa 32000, Israel. [2]Department of Biotechnology and Food Engineering, Technion—Israel Institute of Technology, Haifa 32000, Israel. [3]The Russell Berrie Nanotechnology Institute, Technion—Israel Institute of Technology, Haifa 32000, Israel. ✉e-mail: roeeamit@technion.ac.il

Hairpin forming RNA sequences are widespread in the RNA world and are not strictly associated with disease phenotypes. Such sequences are commonly used in synthetic systems for biological research. Perhaps the most ubiquitous system is composed of RNA sequences that encode for multiple hairpin motifs that can bind the phage coat proteins (CPs) of PP7 or MS2. Using this system to label the 5′ or 3′ end of a transcript has become commonplace in the last two decades[13–18], and enables visualization of RNA transcripts when the CPs are co-expressed. This approach, originally introduced by Singer and others[13–15], was devised for the purpose of probing the dynamics of transcription and other RNA-related processes, irrespective of cell-type. When co-expressed, the coat-protein-bound RNA molecules yield bright puncta, which are similar in appearance to natural biomolecular condensates. Consequently, we hypothesized that co-suspension of synthetic RNA hairpin cassettes together with their binding CPs can lead to the formation of gel-like particles via liquid-solid phase separation in vitro. In addition, by utilizing the CP binding ability of the hairpins, we expect to be able to selectively incorporate proteins of our choosing into the solid-like granules, resulting in a selective platform for the stable concentration of proteins.

In this paper, we rely on our previous works[19–21] to design and synthesize a variety of PP7 coat-protein (PCP) binding synthetic long non-coding RNA molecules (slncRNAs). Using fluorescent RNA nucleotides, we show that these slncRNAs form isolated puncta in vitro in a manner dependent on the number of hairpins encoded into the RNA. We further show that addition of fluorescent PCP to the suspension results in almost complete co-localization between protein and slncRNA. By tracking puncta fluorescence signals over time, we demonstrate that for all slncRNAs used, the various puncta emitted similar signals characterized by bursts of increasing or decreasing fluorescent intensity. We further show that signal intensities and temporal characteristics are dependent on the number of hairpins present in the RNA. Using these observations, we conclude that these "fluorescence-bursts" corresponded to addition or shedding of slncRNA-PCP nucleoprotein complexes. These events occur at rates that are consistent with the puncta being phase-separated solid-like granules. Consequently, we present these slncRNA-protein granules as a genetically encoded platform for the selective storage of proteins as well as a model system for exploration of liquid-solid phase separation.

## Results

### Hairpin containing RNA phase separates in vitro into gel-like granules

To test whether hairpin containing RNA can phase separate in vitro we designed six synthetic long non-coding RNA (slncRNA) binding-site cassettes using our binding site resource[19–21]. We divided our slncRNAs into two groups. For the first group (class I slncRNAs), we designed three cassettes consisting of three, four, or eight hairpins that encode for PCP binding sites (PCP-3x, PCP-4x, and PCP-8x, respectively). In this group, hairpins were spaced by a randomized sequence that did not encode for a particular structure. For the second group (class II slncRNAs), we encoded three cassettes that consisted of three, four, and fourteen PCP binding sites that were each spaced by hairpin structures that do not bind PCP (PCP-3x/MCP-3x, PCP-4x/MCP-4x, and PCP-14x/MCP-15x, respectively). In addition, we designed a negative control slncRNA which does not contain any hairpin binding sites. To ensure that the negative control sequence has a similar GC content as the other slncRNA molecules (45%), it was designed as a permutation of the PCP-8x sequence. The sequences encoding for the slncRNAs were cloned downstream to a pT7 promoter and transcribed in vitro to generate the corresponding RNA. To visualize the RNA, we incorporated fluorescent nucleotides in the transcription reaction such that an estimated 35% of uracil bases were tagged by Atto-488 fluorescent dye. Each slncRNA-type was separately mixed with granule forming buffer (see methods: In vitro granule preparation, and Fig. 1a) at equal

concentration (8.5 nM final concentration) and incubated for 1 h at room temperature. 2–5 µl of the granule reaction were then deposited on a glass slide and imaged using an epi-fluorescent microscope.

The images show that reactions with slncRNA molecules which contain hairpin binding sites result in the formation of a multitude of bright localized fluorescent condensates, with the exception of the PCP-3x case (Fig. 1b). Interestingly, when increasing the slncRNA concentration in the PCP-3x case, sporadic granules do begin to form at 20 nM, and more robustly at 40 nM (Supplementary Fig. 1), reminiscent of a concentration dependent, liquid-like phase separation rather than gel-like. In contrast to the above, the granule reaction containing the negative control slncRNA does not result in any discernible puncta (Supplementary Fig. 2).

In addition to the localized, small condensates, we note the formation of larger structures in the reactions prepared with the longer slncRNA molecules (e.g., PCP-8x and PCP-14x/MCP-15x), consistent with a gel like solid network (Fig. 1c). We examined the median condensate fluorescence obtained per slncRNA sequence. To get a standardized measurement, we normalized the measured fluorescence values by the number of estimated labeled uracil bases in each sequence (assuming a 35% labeling efficacy as reported by the manufacturer). The standardized quantity is then dependent on the number of molecules in a condensate, as well as on the average fluorescence of a single uracil Atto-488 label. The results (Fig. 1d) reveal a dependence on slncRNA class, where class I slncRNA molecules yield condensates with weaker fluorescence when compared with class II. The exception to this is PCP-14x/MCP-15x which appears to be weakest on average. This can be due to reduced fluorescence of a single uracil label brought about by quenching (reduction in fluorescence is estimated to be 4.2x – see methods: Granule microscopy and Fig. 2d). We further examined the background fluorescence from each slncRNA granule reaction and found roughly similar background levels for all slncRNAs (350–450 [A.U]). Normalization of the background (under the assumption that the main contribution to the background is free floating slncRNAs), reveals a dependence on size with the shorter slncRNAs (PCP-4x and PCP-3x/MCP-3x) showing higher normalized fluorescence compared to the negative control (Supplementary Fig. 3), and longer slncRNAs showing weaker values.

Quantification of the area of the localized condensates shows that granule size is generally dependent on slncRNA length rather than on number of hairpins, with shorter slncRNAs resulting in smaller granules (Fig. 1e).

To further analyze the condensate structure, we fitted the normalized condensate fluorescence intensity distributions to a modified Poisson distribution (see Fig. 1f, Supplementary Fig. 4 and methods: Estimating the signal amount per slncRNA-RBP complex). The panels reveal three characteristic distributions. For PCP-4x, an exponential distribution is recorded (i.e., $\lambda = 0$). For PCP-3x/MCP-3x, and PCP-4x/MCP-4x, a Poisson distribution of $\lambda$-1 seems to be the best fit. Finally, for PCP-8x, and PCP-14x/MCP-15x, a Poisson distribution of $\lambda$-2-3 fit best. These results are consistent with the formation of condensates that are characterized by an increasing number of slncRNA molecules that are cross-linked to form a gel-like "granule", where the number of hairpins encoded into the slncRNA determines the average number of molecules or cross-links within the observed field of granules. Moreover, the interpretation suggested by the shape of the distribution is contrasted by the counter-intuitive observation of decreasing value of the of the fitting parameter $K_O$ as a function of an increasing number of hairpins (Fig. 1g). In this particular context, this observation is manifested by a significantly more gradual increase in mean or median granule fluorescence as compared to what would be naively expected by a simple rescale that takes into account the number of hairpins. Together, these observations suggest that slncRNA granules form via cross-linking interaction of multiple slncRNA molecules, and that an increasing number of hairpins and cross-links lead to a denser

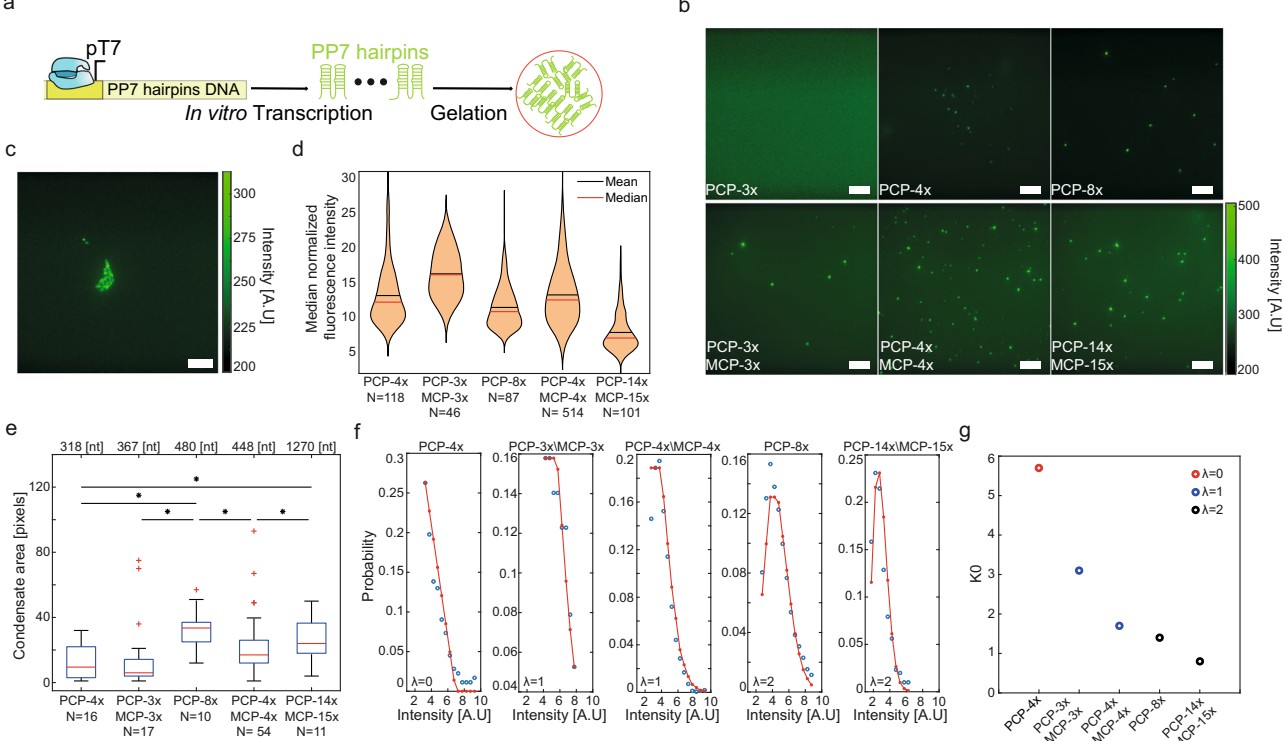

**Fig. 1 | Hairpin-containing slncRNA molecules phase separates in vitro.**
**a** Construct diagram depicting in vitro transcription of hairpin containing slncRNA molecules used and their gelation under suitable conditions. **b** Microscopy images showing dependence of structure morphology on the number of binding sites in the slncRNA. PP7-3x results in no visible puncta, while other slncRNAs shows multiple isolated puncta and additional larger fluorescent structures. All scale bars are 10 μm. **c** Sample image of a PCP-8x granule reaction depicting a larger structure. Scale bar is 10 μm. **d** Violin plots of median condensate fluorescence of slncRNA-only condensates. **e** Boxplots showing area (in pixels) of localized condensates, showing that longer slncRNA molecules typically result in larger condensates. Asterisks denote statistical significance at the 5% level according to a two-sample *t*

test. (*p*-values: PCP-4x vs. PCP-8x: 4e-4; PCP-4x vs. PCP-14x/MCP-14x: 0.046; PCP-4x vs. PCP-3x/MCP-3x: 0.043; PCP-8x vs. PCP-4x/MCP-4x: 0.025; PCP-4x/MCP-4x vs. PCP-14x/MCP-14x: 0.031). On each box, the central mark indicates the median, and the bottom and top edges of the box indicate the 25th and 75th percentiles, respectively. The value for 'Whisker' corresponds to ±1.5 IQR (interquartile rate) and extends to the adjacent value, which is the most extreme data value that is not an outlier. The outliers are plotted individually as plus signs. **f** Poisson function fits for the median fluorescence intensities of the slncRNA granules. **g** $K_O$ estimates calculated from the Poisson fits, showing a dependence on the number of binding sites in the slncRNA molecule. Source data are provided as a Source data file.

condensate. Denser granules, in turn, may result in fluorescence quenching of the labeled uracils[22] leading only to a gradual and disproportionate increase in fluorescence observed (reduction in fluorescence ranging from 1.3x to 4.2x depending on cassette - see Fig. 2d for details).

### RNA-based granules co-localize with protein-binding partners

To test if the hairpins retain their ability to bind the PP7 phage coat protein while in the granule state, we added recombinant tandem dimer PP7 coat protein fused to mCherry (tdPCP-mCherry) to the granule formation reaction in large excess (reactions were set up with 10–20 nM slncRNA concentration and 800 nM protein concentration, see methods: In vitro granule preparation) to saturate the slncRNA molecules while accounting for the multiple binding sites present on one slncRNA molecule (Fig. 2a). The tdPCP-mCherry version used lacks the necessary moiety to form the wildtype viral capsid[23]. The images (Fig. 2b and Supplementary Fig. 5) show colocalization between the 488 nm channel (Atto-488) and the 585 nm channel (mCherry) for all slncRNA designs used in the experiment implying that PP7 coat proteins are able to bind the RNA hairpins in the condensate state. Hence, the slncRNA and their protein partners form synthetic RNA-protein (SRNP) granules. Unexpectedly, PP7-3x granules were witnessed in the presence of the protein. This could indicate that the addition of a protein element adds a measure of multivalency, thus allowing RNA molecules to condensate at a lower

concentration compared to the slncRNA-only case. Another possible explanation could be that three adjacent PP7 stem loops are able to co-localize a sufficient number of tdPCP-mCherry proteins to form a bright punctum which than attracts additional slncRNA molecules (i.e., the protein is the nucleating agent instead of the slncRNA). To check that this condensation was hairpin dependent, we tested whether the control RNA (of the same length and GC content as PCP-8x) containing no designed hairpins, condenses either on its own or in the presence of tdPCP-mCherry. In both cases, no condensates were detected in either the 488 nm or 585 nm channels (Supplementary Fig. 6). Finally, unlike for the slncRNA only case, SRNP granules (particularly for high number of hairpins) show an increased propensity to form large-scale extended structures, suggesting a more complex structure formation and condensation for the SRNP granules as compared with the slncRNA-only case.

To check that the observed colocalization is due to affinity between the coat protein and the corresponding binding sites, we repeated the granule formation reactions with a tandem dimer MS2 coat protein fused to mCherry (tdMCP-mCherry) protein. In this case, granules formed with class I slncRNAs showed no colocalization with the protein, while those formed with class II slncRNAs (which contain MCP binding sites) were colocalized (Supplementary Fig. 7).

Next, we measured the median fluorescence intensity of the tdPCP-mCherry protein in different SRNP granules. The distributions of median values (Fig. 2c) show a clear dependence on the number of

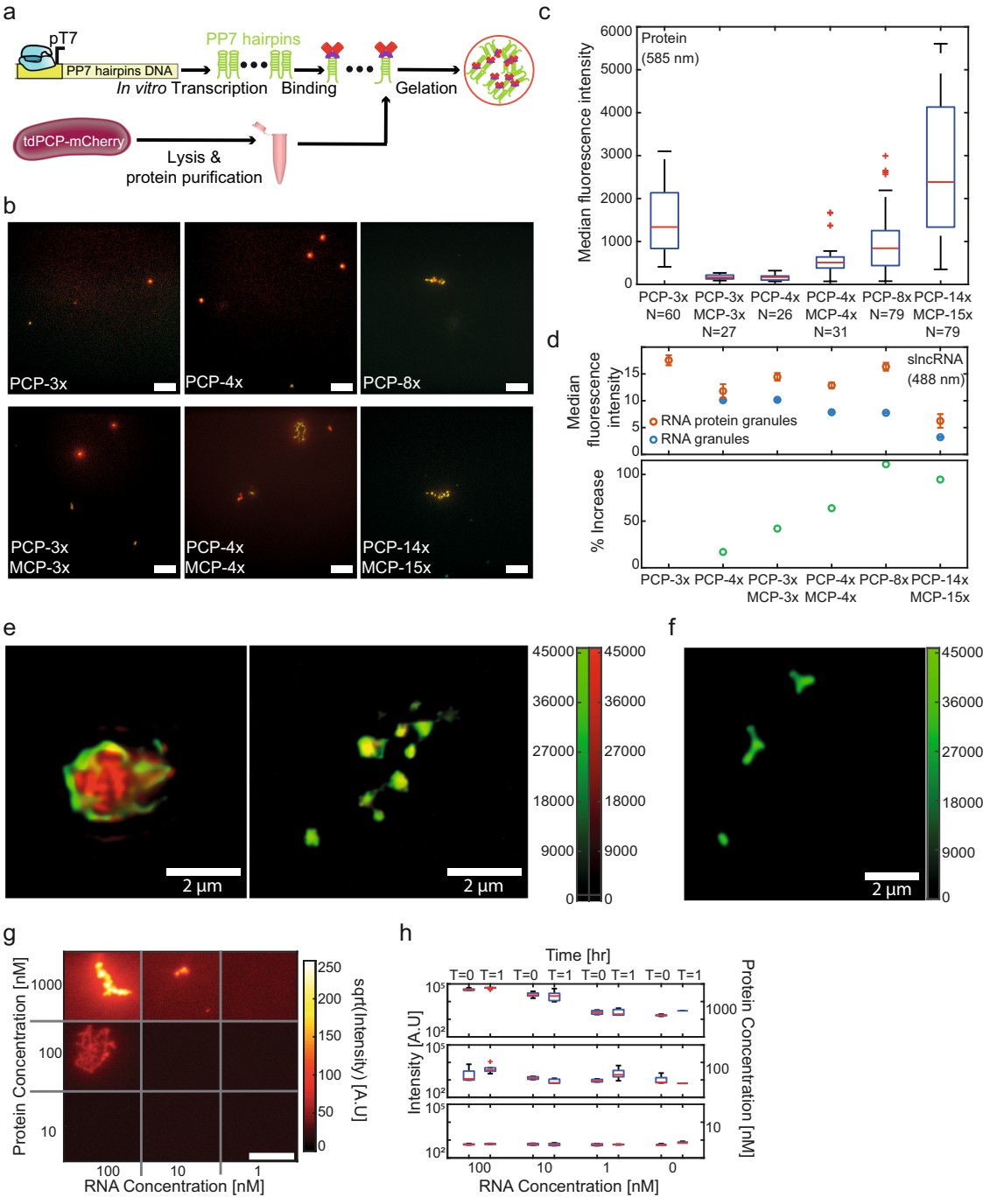

binding sites available for protein binding. First, the PCP-3x/MCP-3x and PCP-4x granules appear to have a similar number of proteins in the granule and are both weaker than PCP-4x/MCP-4x granules, suggesting that PCP-4x slncRNAs inside the granules are not fully occupied by proteins. In addition, the PCP-14x/MCP-15x granules seems to be >2-fold brighter as compared with the PCP-8x granules, despite having <2-fold the number of hairpins. This stands in contrast to the observation that PCP-14x/MCP-15x granules appear to be ~3 times brighter than PCP-4x/MCP-4x granules, reflecting the difference in the number of binding sites available for binding. Finally, PCP-3x granules appear to be half as bright as PCP-14x/MCP-15x granules, providing more evidence that the former are not RNA-dependent entities. We also observe that when the spacing regions within the slncRNA encode for the MCP hairpins, the formed granules contain a larger protein cargo (Fig. 2c).

To confirm this observation, we also observed the SRNP granules in the 488 nm channel. Here a slightly more complex image emerges, whereby the median normalized fluorescence values for the RNA granules decline as the number of hairpins increases, hinting once again at a quenching process due to the tightly packed nature of the condensates. The reduction in fluorescence is estimated to be from 1.3x to 4.2x as a function of slncRNA size due to the increasingly packed nature of the condensates. In contrast, measurements of RNA-protein granules reveal an increase in fluorescence which is proportional to the number of binding sites available for protein binding. This trend peaks at PCP-8x before the effects of quenching become more dominant for the PCP-14x/MCP-15x (estimated to be 2x and 4.2x for the SRNP and RNA-only granules respectively). This behavior indicates at the existence of an optimum point for slncRNA design in terms of number of binding sites and complexity of the design. (Fig. 2d). Together, the

**Fig. 2 | slncRNAs and proteins can form RNA-protein granules in vitro.**
**a** Construct diagram depicting the suspension of tdPCP-mCherry recombinant protein together with in vitro transcribed slncRNA, resulting in synthetic RNA-protein granules. **b** Microscopy images showing an overlay of the 585 nm channel (mCherry) and the 488 nm channel (slncRNA). All scale bars are 10 μm. **c** Boxplots of median 585 nm (mCherry) fluorescence intensity values collected from multiple granules. On each box, the central mark indicates the median, and the bottom and top edges of the box indicate the 25th and 75th percentiles, respectively. The value for 'Whisker' corresponds to ±1.5 IQR (interquartile rate) and extends to the adjacent value, which is the most extreme data value that is not an outlier. The outliers are plotted individually as plus signs. **d** Top - median 488 nm (Atto488) fluorescence intensity values collected from multiple slncRNA granules (blue) and slncRNA-protein granules (orange). Quenching for the slncRNA granules is empirically estimated at 1x for the PCP-4x and PCP-3x/MCP-3x, 1.34x for the PCP-4x/MCP-4x, 1.37 for the PCP-8x, and 4.2x for PCP-14x/MCP-15x. Note that we assume no quenching for the SRNP granules, except for the case of PCP-14x/MCP-15x. Data presented as median values ± SEM. Bottom—Increase in Atto-488 fluorescence between slncRNA-protein granules and slncRNA only granules, for the different slncRNA molecules. Data in top and bottom panels was collected from: 112 PCP-3x/MCP-3x, 165 PCP-4x, 204 PCP-4x/MCP-4x, 121 PCP-8x, and 89 PCP-14x/MCP-15x,

RNA-only granule, and from 91 PCP-3x, 69 PCP-3x/MCP-3x, 30 PCP-4x, 92 PCP-4x/MCP-4x, 85 PCP-8x, and 37 PCP-14x/MCP-15x RNA-protein granules. **e** Structured illumination super resolution images of (left) PCP-14x\MCP-15x slncRNA-protein granule, and (right) PCP-4x slncRNA-protein granules. Color bar indicates fluorescence intensity. **f** Structured illumination super resolution images of PCP-14x/MCP-14x slncRNA-only granules. Scale bar is 2 μm. Color bar indicates fluorescence intensity. **g** Microscopy images for serial dilutions of reaction components taken at T = 1 hr after reaction setup. Highest concentrations show the formation of highly fluorescent filamentous structures, as seen in the top left image. Lower RNA concentrations result in smaller structures, while lower protein concentration result in weaker fluorescence. Scale bar is 10 μm. Due to high dynamic range, the intensities presented are the square root of the raw data images. **h** Maximal observed intensity values for each reaction condition at time T=0 and T=1 hr. All distributions were derived from 5 separate microscopy images of granule reaction prepared with the listed concentrations. On each box, the central mark indicates the median, and the bottom and top edges of the box indicate the 25th and 75th percentiles, respectively. The value for 'Whisker' corresponds to ±1.5 IQR (interquartile rate) and extends to the adjacent value, which is the most extreme data value that is not an outlier. The outliers are plotted individually as plus signs. Source data are provided as a Source data file.

observations in both channels indicate that SRNP granules are less dense gel-like structures as compared with the slncRNA-only granules. To authenticate the granules as being solid-like RNA-protein structures, we imaged them using structured illumination microscopy (SIM) super resolution microscope with 120 nm resolution. Figure 2e-left shows a sample image of a PCP-14x/MCP-15x granule containing the tdPCP-mCherry protein. The image shows that the slncRNA is found mainly in the periphery of the granule, with filaments protruding into its core, where a high amount of protein is amassed in a network like configuration. The RNA seems to encase the protein cargo in a dense shell-like structure. Figure 2e right shows a sample image of PCP-4x granules, depicting cage-like structures with a solid protein core and slncRNA filaments protruding and connecting the different structures. RNA-only granules on the other hand appear to be more compact and uniform in nature, akin to solids (Fig. 2f).

Finally, we explored the phase space of SRNP granule formation. To do so, we characterized formation of the PCP-14x/MCP-15x SRNP granules as a function of both slncRNA and protein concentration. For this we produced non-fluorescent RNA molecules (for higher concentrations) and mixed different titers of slncRNA and tdPCP-mCherry protein, each varied over two orders of magnitude. Puncta like structures were detected only for slncRNA and proteins concentrations of 10 nM and 100 nM respectively, or above (Fig. 2g). The images display bright puncta that are embedded within a filamentous structure. Quantification of the maximal intensity of the puncta both at time T = 0 (i.e., beginning of the reaction) and time T = 1 [hr] (Fig. 2h) reveals a fluorescent intensity distribution which declines by two orders of magnitude (i.e., from ~$10^5$ to ~$10^3$) in a step-like function as the RNA concentration is reduced from 100 to 1 nM, providing further indication that RNA is essential for granule formation. Likewise, the intensity distribution of the puncta declines in a more gradual fashion as the protein concentration is reduced, but overall, a similar disappearance of puncta is observed.

### Dynamic signal analysis of SRNP granules reveals a structure with gel-like characteristics
A hallmark of liquid-liquid phase separation is the exchange of molecules between the dilute phase and the dense phase. This is also true for gels with non-permanent intermolecular interactions, wherein random breaks and rearrangement of the connections which form the inner network allow macromolecules (monomers and small polymers) to diffuse in and out of the gel phase[8–11], albeit at a significantly slower rate as compared with a high density liquid phase. These exchange events are predicted to occur independently of one another, at a rate which depends on multiple parameters: the probability of cross linking

within the gel network (i.e., number of hairpins), the transient concentration of the molecules in the surrounding solution, and the average diffusion rate of the monomers. The movement of molecules (fluorescent CPs, slncRNA, and CP-bound slncRNA complexes) between the different phases should be reflected by changes in granule fluorescence intensity.

To test whether the synthetic granules display this characteristic, we tracked the fluorescence intensity in both the 488 nm channel (for slncRNA), and the 585 nm channel (for protein), of each granule in a given field-of-view for 60 min. We analyzed the brightness of each granule at every time point using a customized analysis algorithm (see methods: Signal analysis and Identifying burst events). The resulting signals are either decreasing or increasing in overall intensity, and dispersed within them are sharp variations in brightness, that are also either increasing or decreasing. Next, we employed a statistical threshold which flagged these signal variation events, whose amplitude was determined to not be part of the underlying signal distribution (p-value<1e-3) (See methods: Identifying burst events, and Supplementary Notes 1–3). We term the statistically significant signal variation events as "signal bursts". These were classified as either increasing bursts (green) or decreasing bursts (red). In addition, we mark the non-significant segments (blue), which are segments where molecular movement cannot be discerned from the noise (Fig. 3a). For each detected burst, we measure its amplitude (Δ intensity) and duration (Δ time), in addition to measuring the time between bursts and the order of their appearance. In Fig. 3b we plot the distributions of amplitudes for all three event types, obtained from ~156 signal traces, each gathered from a different granule composed of PCP-14x/MCP-15x and tdPCP-mCherry. We observe a bias towards negative burst or shedding events. Assuming an interpretation that fluorescent burst events correspond to entry and shedding events of slncRNA-CP complexes into or out of the synthetic granules, the amplitude bias towards negative events is consistent with RNA degradation and lack of transcription within the in vitro suspension, leading to a net shedding of slncRNA-protein complexes out of the granules over time.

To confirm that we are observing entry and shedding events of what are likely single slncRNA molecules into and out of the fluorescent granules vis-a-vis the signal bursts, we tracked the intensity of PCP-14x/MCP-15x RNA-only granules with and without the presence of RNase A. We first found that at enzyme concentrations above 35 nM, no granules were observed whatsoever, indicating correct activity of the enzyme. At a concentration of 35 nM we were able to track identified granules for at least 60 min. Figure 3c depicts a typical signal of a granule in the presence of RNase, showing a steady decline over several minutes. In addition, while shedding events seem to maintain their

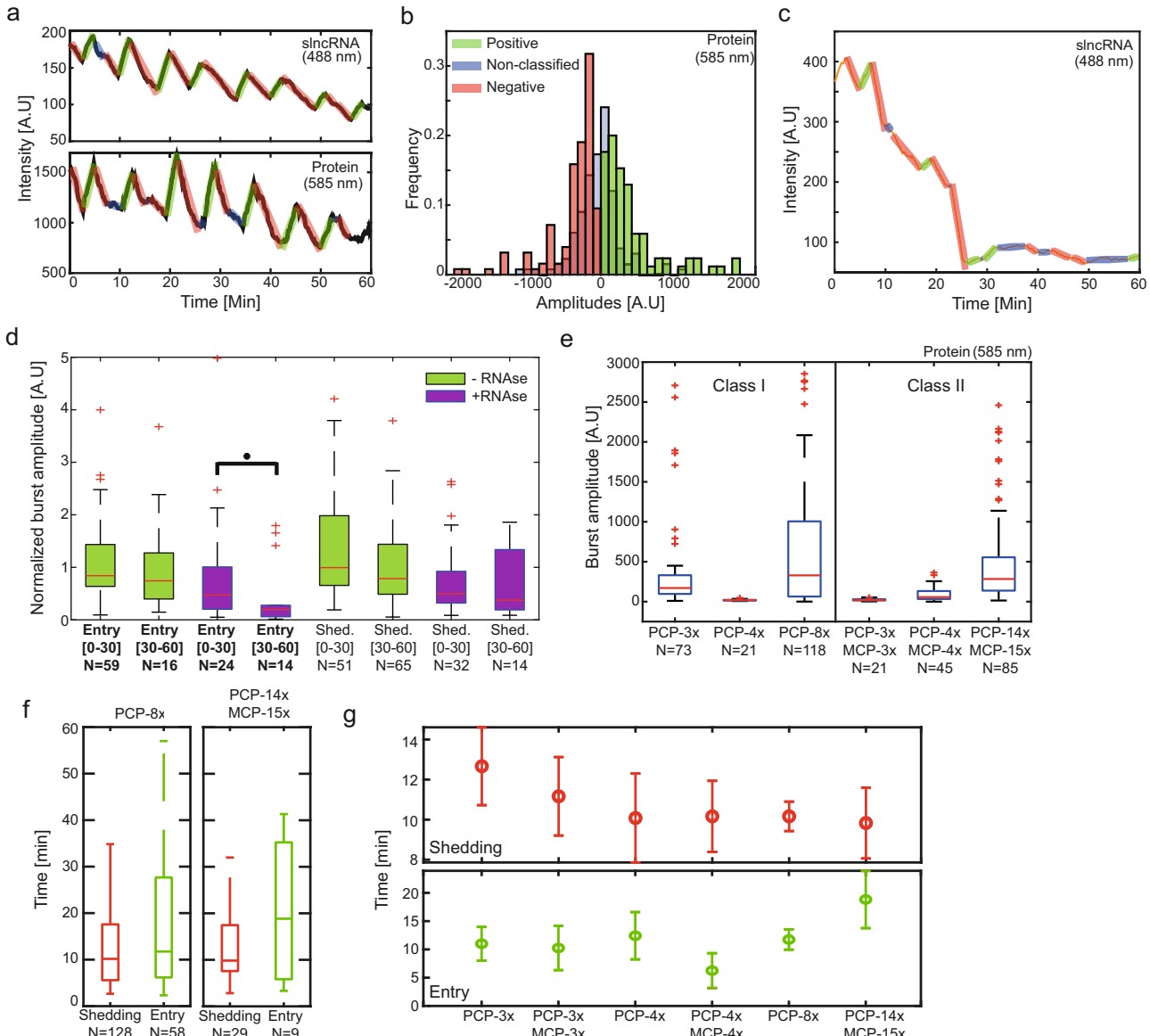

**Fig. 3 | in vitro dynamic signal analysis. a** Matching sample traces of both slncRNA fluorescence (top) and protein fluorescence (bottom) measured from a single granule over the course of 60 min. Signals are overlaid with annotations of puncta signal. Annotations represent increasing intensity burst events (green), decreasing intensity burst events (red), and non-classified signal (blue), respectively. **b** Amplitude distributions gathered from ~156 signal traces in vitro, each collected from a PCP-14x/MCP-15x granule. **c** Sample signal obtained from the tracking a PCP-14x/MCP-15x granules in the presence of 35 nM RNase A enzyme. **d** Boxplots showing burst amplitudes measured from PCP-14x/MCP-15x RNA-only granules with and without the presence of RNase A. Amplitudes are normalized by the estimated number of labeled uracil bases in the slncRNA molecule. Asterisk denotes statistical significance at the 5% level according to a Wilcoxon rank sum test (*p*-value 0.0139). **e** Boxplots depicting positive amplitude distributions for all slncRNAs.

**f** Sample boxplots depicting distributions of durations between a positive burst and a subsequent positive burst (green), and durations between a negative burst and a subsequent negative burst (red). In (**d**–**f**) On each box, the central mark indicates the median, and the bottom and top edges of the box indicate the 25th and 75th percentiles, respectively. The value for 'Whisker' corresponds to ±1.5 IQR (inter-quartile rate) and extends to the adjacent value, which is the most extreme data value that is not an outlier. The outliers are plotted individually as plus signs. **g** Top −median time between successive shedding events, (*N* = 39 PCP-3x, 30 PCP-3x/ MCP-3x, 26 PCP-4x, 45 PCP-4x/MCP-4x, 128 PCP-8x, and 29 PCP-14x/MCP-15x). Bottom−median time between successive entry events, (*N* = 23 PCP-3x, 6 PCP-3x/ MCP-3x, 6 PCP-4x, 18 PCP-4x/MCP-4x, 58 PCP-8x, and 9 PCP-14x/MCP-15x). Data in both panels appears as median values ± SEM. Source data are provided as a Source data file.

amplitude, re-entry events seem to rapidly diminish in amplitude to a median level that is ~10–20% of the original entry burst level (Fig. 3d). In particular, the entry burst amplitude reduced in a statistically significant fashion in the latter part of the tracking (30–60') as compared with the first part of the experiment (0–30' – Wilcoxon rank test *p*-value<0.01). Together, this indicates that slncRNA degradation occurs outside of the granules, while inside the structure they seem to be protected from degradation, consistent with a gel-like phase.

We then proceeded to gather statistical tracking data for granules produced from all previously described slncRNA designs (including the PP7-3x which does not phase separate on its own at our working concentration). Comparison of the amplitude distributions per design (class I vs. II), (Fig. 3e) reveals a dependence on the number of hairpins available for protein binding, where more protein binding sites translate directly into larger amplitudes. As before, PCP-3x is revealed to be an outlier in this case, presenting amplitudes akin to those observed in

PCP-4x/MCP-4x granules, providing another indication of a different phase behavior.

In addition, we measured the time duration between events for each granule type. The observed rate (-10 min or more) is two orders of magnitude above the typical rate observed in liquid phase separated condensates[24], but is in line with the measurements performed on RNA gels by Vale et. al.[11], providing additional confirmation that the SRNP granules are gel-like particles (Fig. 3f). Examination of the median time between bursts reveals that shedding events (negative bursts) occur roughly every 10 min, regardless of slncRNA design and number of binding sites, indicating a global behavior of the formed granules (Fig. 3g-top). Entry events (positive bursts) on the other hand, appear to demonstrate some dependence on the number of binding sites available for protein binding, at least for the slncRNAs with four or more binding sites. For these, the average time between events rises, signifying a reduced ability of bound slncRNA molecules to enter the granules (Fig. 3g-bottom). Such a behavior could indicate saturation of the granule, or a high degree of entanglement in the internal granule structure, hindering entrance of new molecules, while also allowing the stochastic shedding of molecules from the periphery of the granule.

### Competition experiments reveal slow granule equilibration times

We next carried out protein exchange or competition experiments on PCP-14x/MCP-15x granules and measured the dynamics and time-scales associated with protein replacement. To do so we initially prepared PCP-14x/MCP-15x granules with tdPCP-mcherry and allowed the system to equilibrate. Then, immediately prior to imaging, we added tdPCP-mCerulean and observed the granule dynamics in all three channels (i.e., 405 nm for tdPCP-mCerulean, 488 nm for slncRNA, and 585 nm for tdPCP-mCherry – see methods: In vitro granule preparation). Altogether, we tracked 39 co-labeled granules in multiple experiments. Tracking of the formed granules in both channels (405 nm for cerulean, and 585 nm for mCherry) reveals a variety of signals, indicating various types of mixing of the two labels. For instance, we observed events of apparent displacement where one signal increases as the other decreases (Fig. 4a-right) indicating a replacement of the mCherry by mCerulean within the granules. Other signals showed a relatively synchronized signal indicating equilibration of the two protein labels at least within the solution (Fig. 4a. – left). We next computed the Pearson correlation for each pair of signals and plotted the distribution of correlation coefficients in Fig. 4b. The correlation computation shows that while a synchronized signal can be detected in 24 of the 39 signals (R > 0.8), an anticorrelated or unsynchronized signal is still detected in a significant number of the pairs (15 of 39 signals R < 0.8) indicating that a significant percentage of the granules are not found in equilibrium.

Comparing the total number of events in both channels during the first 20' reveals that while the number of entry events is similar (Fig. 4c - N = 25 for mCherry and N = 36 for mCerulean), the number of shedding events is significantly lower in the mCerulean channel (Fig. 4d. N = 40 and N = 2 for the mCherry and mCerulean channels respectively). At longer tracking durations (20–60'), more mCerulean shedding events are observed (N = 23), but are nevertheless x3 lower than the total number of mCherry shedding events (N = 69) over the same duration. Together, the significantly smaller number of shedding events over the one-hour tracking period in the mCerulean channel (N = 109 for mCherry vs N = 25 for mCerulean) as compared with the relatively equal number of entry events (N = 69 for mCherry vs N = 63 for mCerulean) indicates that while the unbound protein equilibrated as expected, the contents of the granules are still not in equilibrium after 1 hr. This interpretation is supported by time interval measurements between successive burst events (Fig. 4e). This analysis reveals a similar rate of entry of tdPCP-cerulean and tdPCP-mCherry proteins

into the granules (Fig. 4e-left), while a discrepancy between the two rates is observed for the shedding events (Fig. 4e-right) in both the duration and total amount of events (N = 73 for tdPCP-mCherry and N = 4 for tdPCP-mCerulean, respectively).

We next examined the burst amplitudes as a function of tracking duration intervals (Fig. 4f–g) in the mCherry channel. The data reveals that burst amplitudes for both entry and shedding events decrease over time with a time scale of -10 min. Specifically, burst amplitudes for the mCherry channel are higher in the first 20 min as compared with the 20–40 and 40–60 min time window in the latter part of the tracking. This indicates a transition from slncRNAs that are fully occupied by tdPCP-mCherry to ones that are increasingly dominated by the tdPCP-mCerulean consistent with rapid equilibration of the proteins within buffer. Consequently, the equilibration of the tdPCP-mCherry and tdPCP-mCerulean within the solution, and the apparent lack of equilibration within the granules at least over the 1 hr duration of the experiments indicated by the unbalanced number of shedding events in both channels, provides additional evidence for a liquid-gel phase transition associated with the SRNP granulation process.

### SRNP granules function as protein capacitors

To provide a measure for the number of slncRNA molecules within the granules, we computed the ratio between the mean granule fluorescence and the mean burst amplitude, assuming the average burst corresponds to one slncRNA molecule bound by proteins. The results (Fig. 5a, b) show that type I gel-like granules (PCP-4x and PCP-8x) have a smaller median number of slncRNAs (-5), as compared with type II gel-like granules (-8–10), suggesting that type II granules form better crosslinked structures. For the PCP-3x/MCP-3x, PCP-4x/MCP-4x, and PCP-14x/MCP-15x the ratio in the red channel displays a dependence on the number of hairpins supporting a more robust solid-like behavior when compared with the type I SRNP granules. We next calculated the "net rate of slncRNA loss", defined as the difference between the total number of observed shedding and entry events, divided by the number of tracked granules (wherein one granule constitutes one hour of tracking data). The rates (Fig. 5c) show a difference between class I and class II granules, as well. While the net loss rate for class I granules increases with number of binding sites, it decreases with the number of binding sites for class II granules. This observation is consistent with type I and type II structures that are characterized by a decreasing and increasing amount of cross-linking, respectively, as a function of the number of hairpins on the slncRNA. Together, this data and the super-resolution microscopy images (Fig. 2e) suggest that class I and II granules form different types of gel-like phases, with the former forming a structure that is permeable to proteins while the latter seem to form robust protein storage nano-particles. In particular, the class II granule characteristics are reminiscent of data and energy storage devices (e.g., capacitors), with the protein cargo replacing the electric charge in the biochemical analog.

To further characterize the "capacitor-like" behavior of the type II granules, we performed a titration experiment with PCP-14x/MCP-15x slncRNAs. We formed granules with a constant slncRNA concentration (120 nM) and different protein concentrations, resulting in a 1:1, 10:1 and 100:1 protein to RNA ratio. We collected shedding burst data for each condition and calculated the previously reported observables. Comparison of the shedding burst amplitudes (Fig. 5d) reveals that granules formed with 10:1 and 100:1 ratios have almost identical burst amplitudes, indicating slncRNA binding saturation. A 1:1 protein to RNA ratio results in amplitudes one tenth the intensity, as expected. Interestingly, the increase in burst amplitude also apparently leads to an increase in the number of slncRNA molecules within the granules. This can be seen from the "duration-between-successive-events" distribution (Fig. 5e), which shows that the time intervals between entry events in the ratio "1" granules are significantly larger than the ratio "10" and "100" intervals (Wilcoxon p-value <0.005). This combined

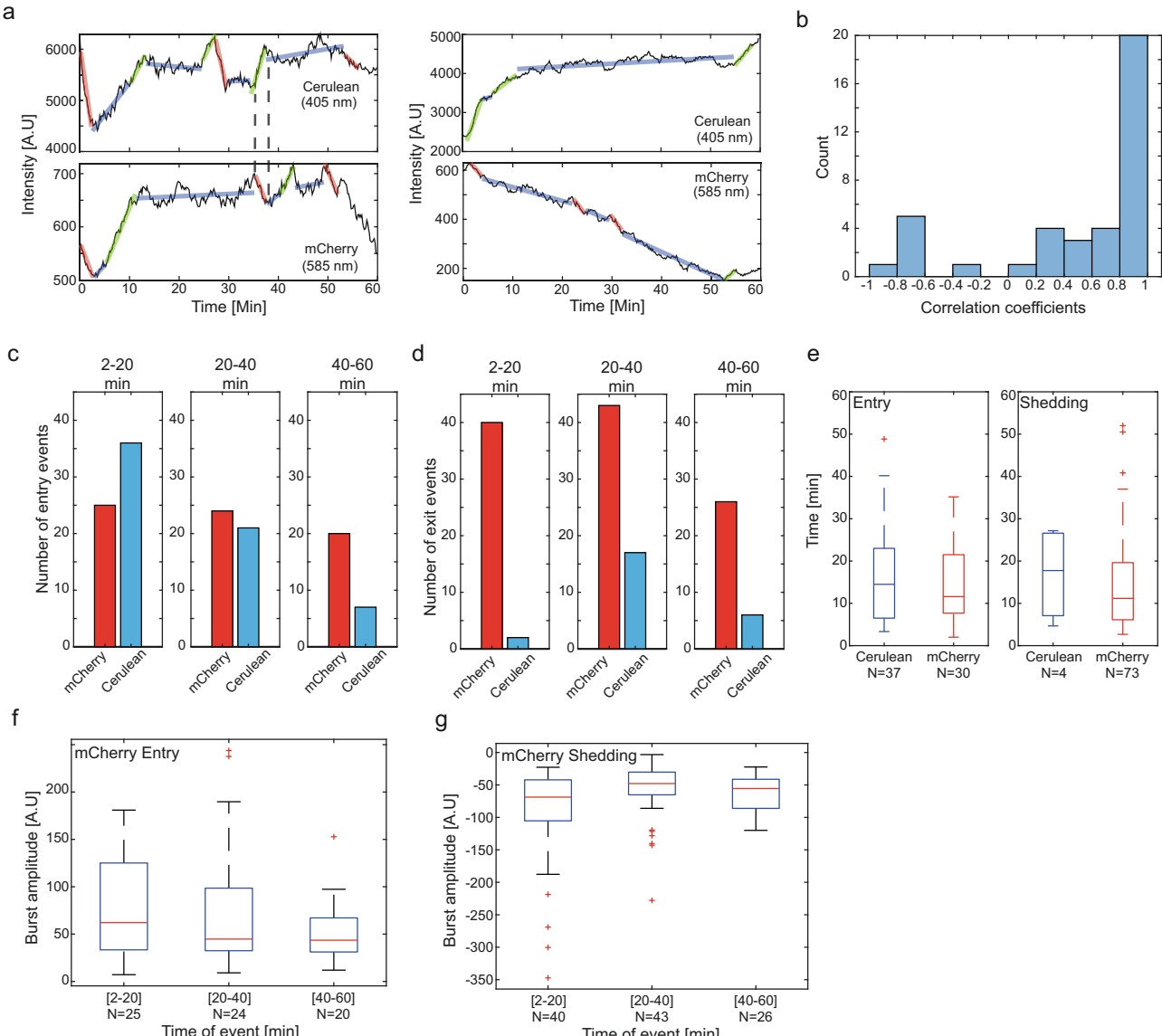

**Fig. 4 | Competition experiments provides support that granulation is a liquid-gel phase transition. a** Sample traces of cerulean fluorescence (top) and mCherry fluorescence (bottom), showing correlation (left) and anti-correlation (right) between the signals. **b** Pearson correlation distribution plot computed for the 39 co-labeled tracks. Number of entry (**c**) and shedding (**d**) events for both channels for different tracking intervals: (left) 2–20', (center) 20–40', and (right) 40–60'. **e** Boxplots depicting distributions of durations between an entry events (left), and durations between shedding events (right), for cerulean fluorescence (blue). Burst amplitudes recorded for entry (**f**) and shedding (**g**) events for the mCherry channel as function of the different tracking intervals: (left) 2–20', (center) 20–40', and (right) 40–60'. In (**e–g**) On each box, the central mark indicates the median, and the bottom and top edges of the box indicate the 25th and 75th percentiles, respectively. The value for 'Whisker' corresponds to ±1.5 IQR (interquartile rate) and extends to the adjacent value, which is the most extreme data value that is not an outlier. The outliers are plotted individually as plus signs. Source data are provided as a Source data file.

with a constant rate of shedding independent of ratio leads to granules that are composed of a smaller number of slncRNAs for the ratio "1" as compared with ratio "10" and "100". Finally, the dependence of the stored protein load on the burst amplitude allows us to define a proportionality constant uniquely for every granule-type. This type of proportionality constant is analogous to electrostatic "capacitance", and can be defined by a biochemical analog to the capacitance equation q=CV. Here, the biochemical "charge" is the total amount of protein stored within the granules, and the protein concentration within the solution corresponds to a biochemical "voltage". Consequently, the protein to slncRNA titration measurement provides a more solid footing for the capacitor analogy, suggesting that a potential protein-storage set of applications may be facilitated both in vitro and in vivo.

## Expression of slncRNAs and protein in bacteria yields puncta-like condensates

Given the capacitor analogy, we hypothesized that in vivo the granules can be used as devices that store granule-bound proteins indefinitely. This is due to the steady state production of slncRNAs and proteins via the cellular transcriptional and translational machinery, that ensures a constant flux of proteins into the granules. To show this, we first proceeded to test whether the granule material characteristics that are measured in vivo match the in vitro measurements. To do so, we decided to utilize two previously reported slncRNA designs which were shown to yield bright localized puncta in vivo in earlier work[19]. The first slncRNA is of a class II design, PCP-4x/ QCP-5x, consisting of four native PCP binding sites and five native Qβ coat protein (QCP) hairpins used as spacers in an interlaced manner. The second slncRNA

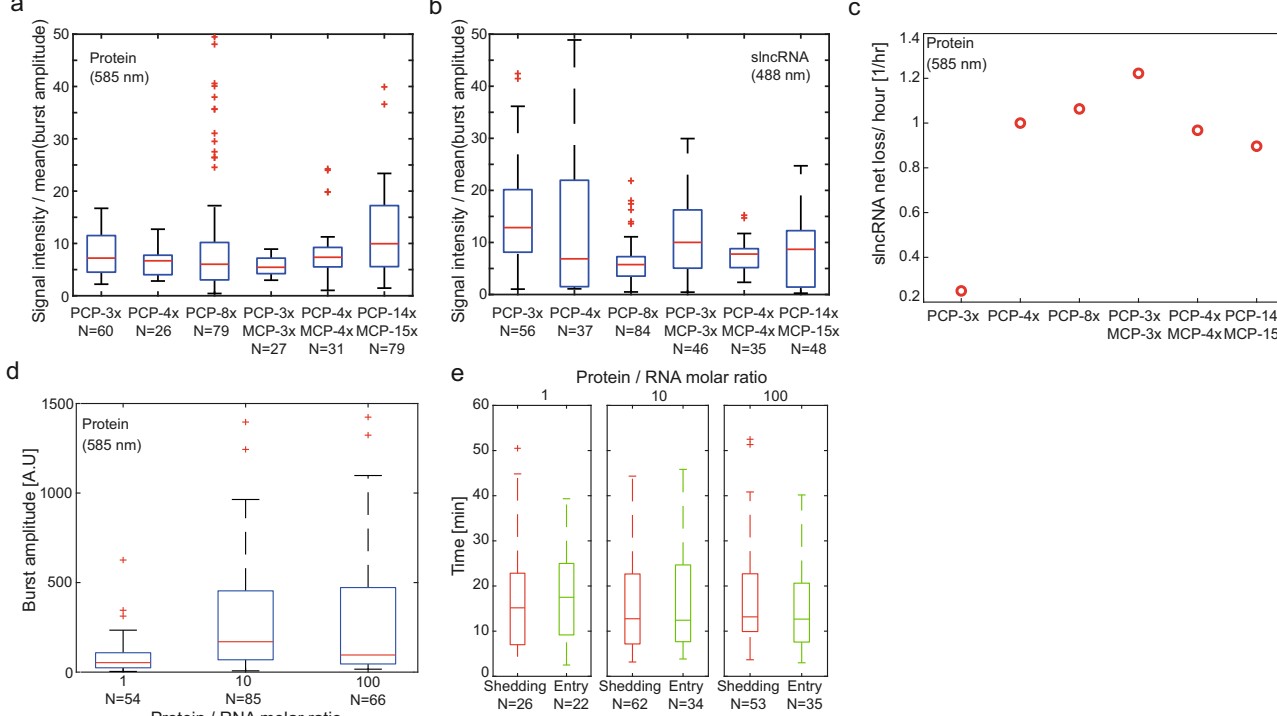

**Fig. 5 | Granules act as protein capacitors. a** Boxplots depicting ratio between granule protein fluorescence and median burst amplitude, providing a measure of the protein content inside the granules. **b** Boxplots depicting ratio between granule slncRNA fluorescence and mean burst amplitude providing a measure of the slncRNA content inside the granules. **c** Rate of net slncRNA loss for each granule type, showing a difference in trend between class I and class II slncRNA. **d** Boxplots depicting burst amplitudes measured from granules with different ratios of protein to slncRNA. Granules formed with ratio of 10:1 and 100:1 appear to have similar amplitudes, indicating slncRNA binding saturation. **e** Boxplots depicting distributions of durations between an entry event (green), and durations between shedding events (red), measured from granules with different ratios of protein to slncRNA. In (**a**, **b**, **d**, **e**) On each box, the central mark indicates the median, and the bottom and top edges of the box indicate the 25th and 75th percentiles, respectively. The value for 'Whisker' corresponds to ±1.5 IQR (interquartile rate) and extends to the adjacent value, which is the most extreme data value that is not an outlier. The outliers are plotted individually as plus signs. Source data are provided as a Source data file.

is the ubiquitous PCP-24x cassette[25], which from the perspective of this work can be regarded as a class I design slncRNA.

To confirm the granules form condensates in vivo, we encoded the slncRNA component under the control of a T7 promoter, and the tdPCP-mCherry under the control of an inducible pRhlR promoter (Fig. 6a). We first wanted to test whether puncta develop in vivo and whether they are dependent on the existence of hairpins in the RNA. For this we co-transformed plasmids encoding either the negative control RNA or the PCP-4x/QCP-5x slncRNA, together with a plasmid encoding for the tdPCP-mCherry protein, into BL21-DE3 *E. coli* cells. Examination of cells expressing the slncRNA and protein following overnight induction of all components revealed the formation of bright puncta at the cell poles (Fig. 6b), which were absent in cells expressing the control RNA which lacks hairpins (Fig. 6c). In addition, the difference between the cultures was even visible to the naked eye (Supplementary Fig. 8), indicating copious amounts of protein which appear to be dependent on number of binding sites encoded in the slncRNA. We believe this phenomenon was missed in the past since such binding sites were exclusively used to track individual mRNA transcripts in vivo where both low concentrations and the effects of translation might hinder the formation of large macro-molecular structures.

Next, to test whether cellular concentration of slncRNA influences the formation of the granules, we quantified the fraction of puncta per cell for cells expressing the PCP-4x/QCP-5x from a multicopy expression vector, and cells expressing the same slncRNA from a bacterial artificial chromosome (BAC) expression vector which is maintained at a single copy level in cells. We found that cells containing the multicopy plasmid frequently present puncta in at

least one of the poles, while cells containing the single copy generally show between zero and one punctum (Fig. 6d). Given that cells expressing the slncRNAs from single copy vectors still present puncta, we decided to continue using this expression vector in follow-up experiments to reduce variability stemming from copy number differences.

We compared cells expressing the PCP-4x/ QCP-5x or the PCP-24x (expressed from a BAC vector) in terms of the spot per cell fraction. Much like in the in vitro experiments, we found a dependence on the number of binding sites in accordance with the in vitro results and the cross linking model of gel phase formation[6,26] (Fig. 6d). Finally, to test whether the polar localization of the granules is a consequence of nucleoid exclusion[27], we grew the cells in starvation conditions for several hours, triggering a transition to stationary phase. In stationary phase the nucleoid is known to condense[28–30], thus increasing the amount of cellular volume which is likely to be molecularly dilute. This, in turn, generates a larger accessible cellular volume for granule formation, which should lead to different presentation of the phase-separation phenomena as compared with exponentially growing cells. In Fig. 6e, we show an image of bacteria displaying 'bridging' (the formation of a high intensity streak between the spots) whereby granules seem to fill out the available dilute volume. This behavior is substantially different than the puncta appearing under normal conditions. Such behavior was observed in >40% of the fluorescent cells and was not detected in non-stationary growth conditions. Thus, SRNP granules with characteristics that are consistent with the in vitro observations form in vivo, in a semi-dilute bacterial cytosolic environment and independent of cell-state.

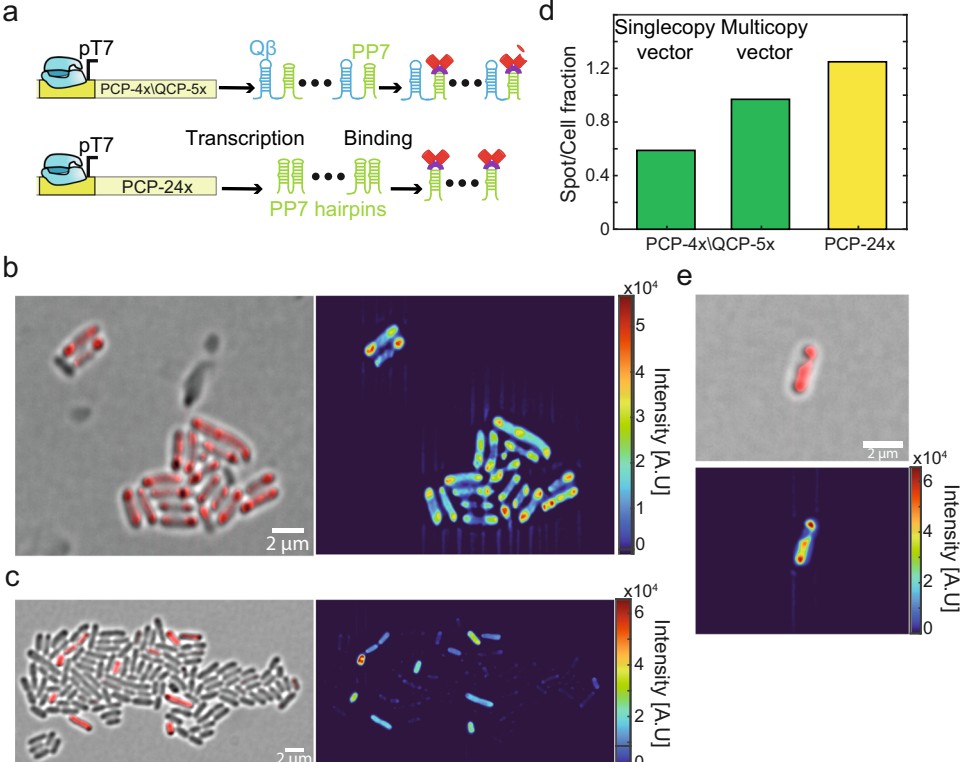

**Fig. 6 | Synthetic phase separated condensates within bacterial cells.**
**a** Construct diagram depicting expression of the two slncRNA cassettes used in the in vivo experiments, in the presence of tdPCP-mCherry. **b** (Left) Merged structured illumination brightfield-585 nm image of cell expressing the PCP-24x slncRNA together with tdPCP-mCherry. (Right) Heatmap of the same image showing a highly fluorescent punctum as cell poles. Color bar indicates fluorescence intensity. **c** (Left) Merged brightfield-585 nm image of cell expressing the negative control RNA together with tdPCP-mCherry. (Right) Heatmap of the same image showing a weak uniform fluorescence across the cell, color bar indicates fluorescence intensity. **d** Bar plot showing fraction of puncta per cell. (Green) PCP-4x/QCP-5x expressed from either a single copy or a multicopy expression vector. (Yellow) PCP-24x expressed from a single copy vector. **e** Typical image of fluorescent bacteria in stationary phase, which are different than the 1-2 puncta image obtained for exponentially growing cells. The presented cell shows "bridging" or spreading of puncta. Bottom image show heatmaps of the top image. All scalebars are 2 μm.

## slncRNA expression increases cellular protein concentration

To investigate the dynamic properties of granules formed in vivo, we utilized the same analysis approach as was used in the in vitro experiments, with minor differences. Normalizing the fluorescence of the granule by that of the cell (see methods: Image analysis and Signal analysis) for every time point results in a signal vs. time trace largely independent from the effects of photobleaching and cellular background noise, allowing us to search for and measure burst events, as was done previously. In Fig. 7a, we plot the distributions of amplitude of all three event types (positive, negative, and non-classified), obtained from 255 traces gathered from cells expressing the PCP-4x/QCP-5x slncRNA together with the tdPCP-mCherry protein. The symmetry in both shape and spread of the negative and positive distributions indicates that both are measurements of the same type of macromolecule, distinguished only by the direction in which it travels (into or out of the granule). Moreover, a similarly symmetric burst distribution is recorded for the PCP-24x slncRNA (Supplementary Fig. 9). This result contrasts with the in vitro amplitude distribution data (Fig. 3b), which presented a skewness towards negative bursts. This implies that in vivo, the transcriptional and translational processes in the cell balance the loss of granule components due to degradation.

Next, we measured the amplitudes of the bursts for both slncRNAs and found that positive and negative amplitudes are proportional to the number of binding sites within the encoded cassette (Fig. 7b). In addition, a more quantitative analysis of these distributions (Supplementary Fig. 10) reveals that a single burst for the 24x cassette is ~2.5-3x more fluorescent as compared with the 4x cassette,

indicating that the molecules transitioning in and out of the 24x granules are slncRNAs partially or fully bound rather than lone proteins. Moreover, estimations of the positive and negative amplitudes are practically equal per slncRNA, providing additional evidence that these are in fact representations of one physical process, with the difference being the directionality of the transitioning slncRNA-protein molecule. Finally, we measured the duration between burst events, revealing that slow shedding and absorption processes on the order of minutes are taking place for the in vivo granules as well (Fig. 7c). Altogether, the non-existence of puncta in cells expressing the negative control RNA, the slow shedding/entry rate of molecules, and the dependence on the number of binding sites, suggest that synthetic RNA protein granules are phase separated condensates in vivo and possess the same gel-like characteristics that were observed for the in vitro suspensions. Consequently, in vivo burst analysis is consistent with the capacitor model, where the amount of protein stored within the SRNP granule seems to be in steady state when there is a steady supply of protein and slncRNA.

Next, to ascertain whether the granules facilitate increased protein titers in vivo in accordance with the capacitor model predictions, we measured for each bright granule the mean fluorescence intensity (Fig. 7d), and the mean intensity of the cell which contains it (Fig. 7e). We observed a dramatic increase in mean cellular fluorescence between cells which express only tdPCP-mCherry and cells which express it together with a slncRNA, suggesting that slncRNA molecules have some effect in the cytosol, regardless of the granules. To quantify this phenomenon more accurately, we measured the total fluorescence of the population using flow cytometry. For this, we grew cells

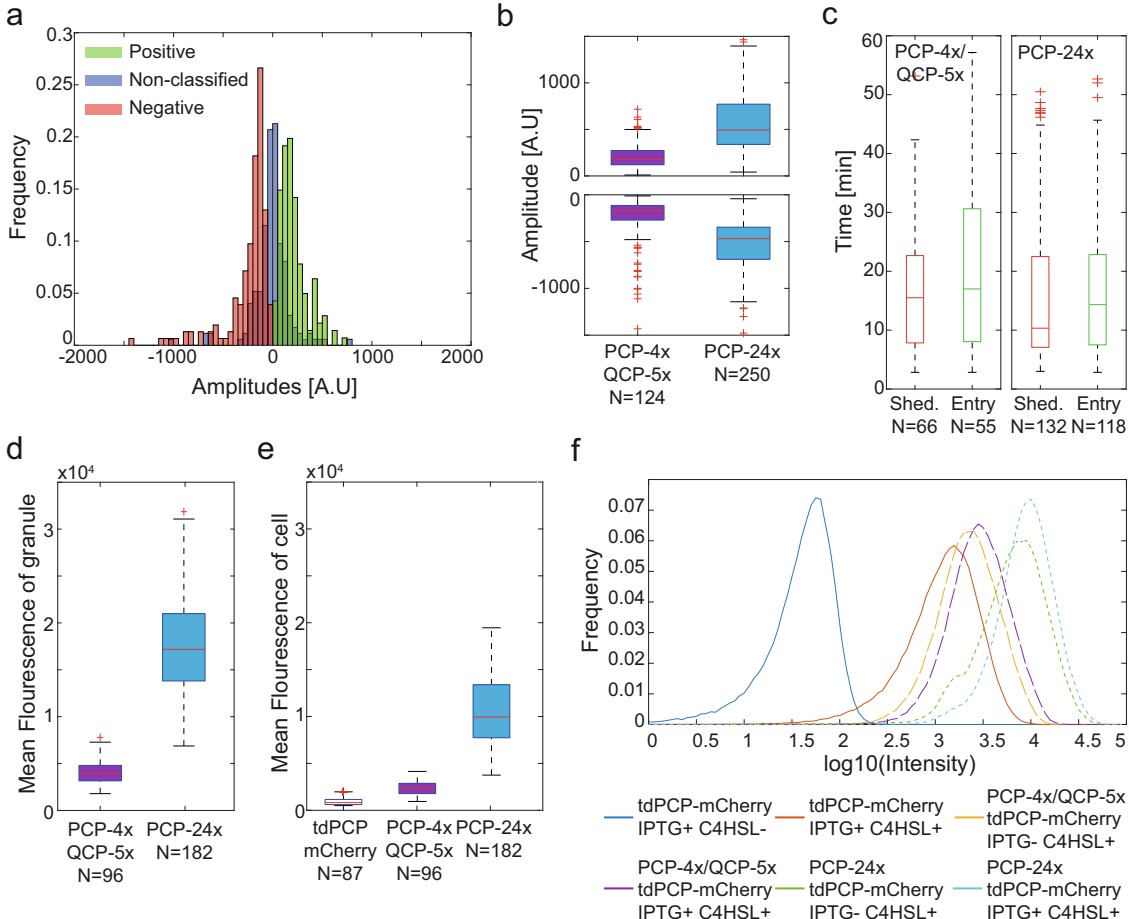

**Fig. 7 | In vivo granules present similar dynamics as in vitro. a** Empirical amplitude distributions gathered from 255 traces in vivo from cells expressing the PCP-4x/QCP-5x slncRNA together with the tdPCP-mCherry protein. **b** Boxplots depicting burst amplitude distributions (top−positive bursts, bottom−negative bursts). **c** Boxplots depicting distributions of durations between entry events (green), and durations between shedding events (red). **d** Boxplot of mean granule fluorescence intensity. **e** Boxplot of mean cell fluorescence intensity. In (**b**−**e**) On each box, the central mark indicates the median, and the bottom and top edges of the box indicate the 25th and 75th percentiles, respectively. The value for 'Whisker' corresponds to ±1.5 IQR (interquartile rate) and extends to the adjacent value, which is the most extreme data value that is not an outlier. The outliers are plotted individually as plus signs. **f** Population intensities of *E. coli* BL21 cells expressing tdPCP-mCherry with different slncRNAs and different combinations of induction as measured by flow cytometry. Source data are provided as a Source data file.

expressing only the protein component (tdPCP-mCherry), and cells expressing both protein and a slncRNA (PCP-4x/QCP-5x or PCP-24x), with different combinations of induction: IPTG (induces the slncRNA) and C4HSL (induces the protein). The data (Fig. 7f and Supplementary Fig. 11) shows that cells expressing a slncRNA, regardless of induction (due to T7 leakiness), show higher fluorescence than cells expressing the protein only. In addition, induction of slncRNA expression with IPTG results in an increase in fluorescence, indicating that slncRNA is a deciding factor in this behavior. Finally, cells expressing the PCP-24x slncRNA show higher fluorescence than cells expressing PCP-4x/QCP-5x, displaying a dependence of the cellular protein titer on number of binding sites available for protein binding.

## Discussion

In this study, we show that synthetic gel-like RNA−protein granules can be designed and assembled using phage coat proteins and RNA molecules that encode multiple CP hairpin binding sites, both in suspension and in vivo. Using fluorescently labeled RNA, we show that granule formation is nucleated by RNA-RNA interactions that are proportional to the number of hairpins encoded into the RNA. In addition, the binding of the proteins seems to further enhance and assist the granule formation process. Using fluorescent single molecule signal analysis, we reveal entry and shedding events of molecules

into and out of the granules. By investigating their size and rate of occurrence, we show that these events correspond to entry and shedding of protein-bound slncRNA molecules, and that they are dependent on the number of hairpins available for protein binding. Transitioning of macro-molecules across a phase boundary is frequently observed in phase-separated condensates, particularly in liquid-liquid based system. In particular, the frequency of these transitions reflects the underlying order, internal interactions, and density of the condensed phase. While in liquid-liquid phase separation systems such transitions occur on the scale of seconds or less, here we observe shedding and insertion events on a much longer time scale of minutes or longer, that is more consistent with a solid or gel-like condensed phase. We provide additional evidence for the liquid-gel phase transition underlying the granulation process, by taking the system out of equilibrium and observing the different equilibration times of the dilute solution and dense granule phases. To do so, we used a binding competitor in one experiment and RNAse in the other. For those experiments, the liquid phase showed rapid mixing in the former and substantial catalysis in the latter, while the granule phase showed slow mixing and undetectable amount of catalysis, respectively. The out-of-equilibrium experiments allowed us to differentiate between the liquid-gel transition from the other options which include liquid-liquid, liquid-glass, and liquid-solid transition. As a result, we

believe that equilibrium characterizations should be considered a standard tool in similar biomolecular phase separation studies.

We further characterized two options for slncRNA design: a homogeneous design which is comprised of multiple CP hairpin binding site and non-structured spacing regions (class I), and a hybrid design which is comprised of hairpin binding sites and additional hairpins in the spacing regions (class II). We show that the design choice has implications for the granule's protein-carrying capacity and dynamics. In particular, class II granules formed particles with increased cross-linking capability in the RNA-only granule, which in turn led to an increased ability to insulate the protein cargo in the SRNP granule phase. On the flip side, class I granules were characterized by decreased cross-linking in the RNA-only phase and increased permeability of the protein cargo in the SRNP-granule phase. In addition, class I granules displayed a faster shedding or dissolution rate, which in turn lead to a smaller protein cargo on average. The slow release and strong internal interactions which keep the granules intact for long durations within a gel-like phase, combined with the selectivity of our system due to the RNA binding component, could be utilized as a programmable controlled release mechanism in suitable biological settings. Hence, our granules can be thought as protein and RNA storage modules akin to a capacitor, with 'capacitance' that is dependent on protein concentration, and a monophasic release profile that can be tuned based on slncRNA design. This two-dimensional phase space of capacity vs. rigidity offers substantial flexibility and tunability when designing SRNP granules for a variety of applications.

The capacitor- or storage-like behavior displayed by the SRNP granules implies that in vivo, the granules together with the gene-expression machinery form a biochemical analog of an RC-circuit. In a conventional RC-circuit, energy is stored within the capacitor for release at a later time. Such circuits are often used to protect electrical devices against sudden surges or stoppages of power. Here, the protein and slncRNA flux into the cytosol correspond to the current, which results in the formation of fully "charged" SRNP granules. This genetically encoded slncRNA and protein storage facility, which is constantly maintained, effectively increases the protein and slncRNA content of the cell beyond the steady-state levels facilitated by standard transcription, translation, RNA degradation, and proteolysis. This storage capacity is precisely the function that is carried out by capacitors in RC-circuits, allowing electrical devices to function even after "power" is cut-off. In the case here, the granules can be used not only to increase levels of a protein of choice by nearly an order of magnitude (as shown in Fig. 7 and Supplementary Fig. 8) without adversely affecting the cell but may also provide a mechanism to increase the cell's ability to survive when a harsh or stressful environment is encountered. While the former may have important implications to the biotechnology sector, the latter may hint at an important function that natural granules (e.g., paraspeckles, p-bodies, etc.) may have evolved for in vivo. In particular, our finding of liquid-gel phase transitions may be also relevant to repeat expansion disorders, which are associated with RNAs containing multiple structural repeats[11]. Further studies will be required to explore the biological relevance of SRNP granules to the survivability of cells and organisms under various forms of stress, and to potential underlying mechanisms for various diseases.

## Methods

### Bacterial strains
*E. coli* BL21-DE3 cells which encode the gene for T7 RNAP downstream from an inducible pLac/Ara promoter were used for all reported experiments. *E. coli* TOP10 (Invitrogen, Life Technologies, Cergy-Pontoise) was used for cloning procedures.

### Addgene plasmids
pCR4-24XPP7SL was a gift from Robert Singer (Addgene plasmid # 31864; http://n2t.net/addgene:31864; RRID: Addgene_31864).

pBAC-lacZ was a gift from Keith Joung (Addgene plasmid # 13422; http://n2t.net/addgene:13422; RRID: Addgene_13422).

### Construction of the slncRNA plasmids
All sequences encoding for the in vitro slncRNAs (i.e., PP7-3x, PP7-4x, PP7-3x/MS2-3x, PP7-4x/MS2-4x, PP7-8x and PP7-14x/MS2-15x. Sequences appear in Supplementary Data 1) were ordered from Integrated DNA Technologies (IDT), (Coralville, Iowa) as gBlock gene fragments downstream to a T7 promoter and flanked by EcoRI (New England Biolabs (NEB),Ipswich, MA, #R3101L) restriction sites on both sides. gBlocks were cloned into a high-copy plasmid containing an Ampicillin resistance gene and verified using Sanger sequencing.

The 5Qβ/4PP7 slncRNA sequence was ordered from GenScript, Inc. (Piscataway, NJ), as part of a puc57 plasmid, flanked by EcoRI and HindIII (NEB, #R3104L) restriction sites. pBAC-lacZ backbone plasmid was obtained from Addgene (plasmid #13422). Both insert and vector were digested using the said restriction enzymes and ligated to form a circular plasmid using T4 DNA ligase (NEB, #M0202L). Sequence was verified by sanger sequencing.

### Design and construction of fusion-RBP plasmids
Fusion-RBP plasmids were constructed as previously reported[21]. Briefly, RBP sequences lacking a stop codon were amplified via PCR off either Addgene or custom-ordered templates. Both RBPs presented (PCP and QCP) were cloned into the RBP plasmid between restriction sites KpnI and AgeI (NEB, catalog: #R3142L and #R3552L respectively), immediately upstream of an mCherry gene lacking a start codon, under the so-called RhlR promoter containing the rhlAB las box[31] and induced by N-butyryl-L-homoserine lactone (C4-HSL) (Cayman Chemicals, Ann Arbor, Michigan. #10007898). The backbone contained either an Ampicillin (Amp) or Kanamycin (Kan) resistance gene, depending on experiment.

### In vitro transcription of slncRNA
A vector containing the slncRNA DNA sequence, flanked by two EcoRI restriction sites, was digested with EcoRI-HF per the manufacturer's instructions to form a linear fragment encoding the slncRNA sequence. The enzyme was then heat-inactivated by incubating the restriction reaction at 65 °C for 20 min. For fluorescently labeled RNA, 1 μg of the restriction product was used as template for in vitro transcription using HighYield T7 Atto488 RNA labeling kit (Jena Bioscience, Jena, Germany, RNT-101-488-S), according to the manufacturer's instructions. Non-fluorescent RNA was transcribed using the HiScribe™ T7 High Yield RNA Synthesis Kit (NEB, E2040S). Following in vitro transcription by either kit, the reaction was diluted to 90 μl and was supplemented with 10 μl DNAse I buffer and 2 μl DNAse I enzyme (NEB #M0303S) and incubated for 15 min at 37 °C to degrade the DNA template. RNA products were purified using Monarch RNA Cleanup Kit (NEB, #T2040S) and stored in −80°.

### Protein expression and purification
*E. coli* cells expressing fusion proteins (tdPCP-mCherry / tdPCP-cerulean / tdMCP-mCherry) were grown overnight in 10 ml LB with appropriate antibiotics at 37 °C with 250 rpm shaking. Following overnight growth, cultures were diluted 1/100 into two vials of 500 ml Terrific Broth (TB: 24 g yeast extract, 20 g tryptone, 4 ml glycerol in 1 L of water, autoclaved, and supplemented with 17 mM KH2PO4 and 72 mM K2HPO4), with appropriate antibiotics and induction (500 μl C4-HSL) and grown in 37 °C and 250 rpm shaking to optical density (OD) > 10. Cells were harvested, resuspended in 30 ml resuspension buffer (50 mM Tris-HCl pH 7.0, 100 mM NaCl and 0.02% NaN3), disrupted by four passages through an EmulsiFlex-C3 homogenizer (Avestin Inc., Ottawa, Canada), and centrifuged (10,000 g for 30 min) to obtain a soluble extract. Fusion protein was purified using HisLink Protein purification resin (Promega (Madison, WI) #V8821) according

to the manufacturer's instructions. Buffer was changed to 1xPBS (Biological Industries, Israel) using multiple washes on Amicon ultra-2 columns (Merck, Burlington, MA #UFC203024).

## In vitro granule preparation

In vitro experiments were performed in granule buffer (final concentrations: 750 mM NaCl, 1 mM MgCl2, 10% PEG4000). Reactions were set up at the appropriate concentrations and allowed to rest at room temperature for 1 h. 3–5 μl from the reaction was then deposited on a glass slide prior to microcopy.

For the RNase experiment, granules were first formed as described and allowed to rest at room temperature for 1 h. Following this, RNase A enzyme (Thermofisher, Waltham, MA, #EN0531) was added at 35 nM final concentration. The reaction was then immediately deposited on a glass slide and proceeded to imaging.

For the competition experiment, granules were first formed with tdPCP-mCherry at a final concentration of 40 nM and allowed to rest at room temperature for 1 h. Following this, tdPCP-mCerulean was added to the reaction at 80 nM final concentration prior to imaging.

## Bacterial culture growth

BL21-DE3 cells expressing the two plasmid system (single copy plasmid containing the binding sites array, and a multicopy plasmid containing the fluorescent protein fused to an RNA binding protein) were grown overnight in 5 ml Luria Broth (LB), at 37° with appropriate antibiotics (Cm, Amp), and in the presence of two inducers – 1.6 μl Isopropyl β-D-1-thiogalactopyranoside (IPTG) (final concentration 1 mM), and 2.5 μl C4-HSL (final concentration 60 μM) to induce expression of T7 RNA polymerase and the RBP-FP, respectively. Overnight culture was diluted 1:50 into 3 ml semi-poor medium consisting of 95% bioassay buffer (BA: for 1 L−0.5 g Tryptone [Bacto], 0.3 ml glycerol, 5.8 g NaCl, 50 ml 1 M MgSO4, 1 ml 10×PBS buffer pH 7.4, 950 ml DDW) and 5% LB with appropriate antibiotics and induced with 1 μl IPTG (final concentration 1 mM) and 1.5 μl C4-HSL (final concentration 60 μM). For stationary phase tests, cells were diluted into 3 ml Dulbecco's phosphate-buffered saline (PBS) with similar concentrations of inducers and antibiotics. Culture was shaken for 3 h at 37° before being applied to a gel slide [3 ml PBSx1, mixed with 0.045 g SeaPlaque low melting Agarose (Lonza, Switzerland), heated for 20 seconds and allowed to cool for 25 min][32]. 1.5 μl cell culture was deposited on a gel slide and allowed to settle for an additional 30 min before imaging.

## Granule microscopy

Granules were imaged in a Nikon Eclipse Ti-E epifluorescent microscope (Nikon, Japan) with a 100×1.45 NA oil immersion objective. Excitation was performed by a CooLED (Andover, UK) PE excitation system at 488 nm (Atto 488) for experiments containing fluorescent RNA, 585 nm for mCherry protein, and 405 nm for cerulean protein. Images were captured using the Andor iXon Ultra EMCCD camera with a 250 msec exposure time for 488 nm, 250 msec exposure time for 585 nm, and 2 seconds for 405 nm. Microscopy control and data acquisition was performed using Nikon NIS-Elements version 4.20.02 (build 988) 64 bit.

Fluorescence quenching for the granules was estimated by first placing each slncRNA sequence on a two-dimensional space, with the x-axis being the estimated number of fluorescent uracil nucleotides (assuming 35% labeling efficiency), and the y-axis being the normalized RNA-only granule fluorescence values as depicted in Fig. 2d top. Estimated non-quenched fluorescence values were extrapolated by fitting a linear line to the data from PCP-4x and PCP-3x/MCP-3x which are assumed to be roughly non-quenched. A numerical value per slncRNA was then calculated by dividing the estimated non-quenched fluorescence by the fluorescence value measured empirically.

## Live cell microscopy

Gel slide was kept at 37° inside an Okolab microscope incubator (Okolab, Italy). A time lapse experiment was carried out by tracking a field of view for 60 min on Nikon Eclipse Ti-E epifluorescent microscope (Nikon, Japan) using an Andor iXon Ultra EMCCD camera at 6 frames-per-minute with a 250 msec exposure time per frame. Excitation was performed at 585 nm (mCherry) wavelength by a CooLED (Andover, UK) PE excitation system. Microscopy control and data acquisition was performed using Nikon NIS-Elements version 4.20.02 (build 988) 64 bit.

Quantification of the fraction of cells presenting puncta was done by taking 10–15 snapshots of different fields of view (FOV) containing cells. The number of cells showing puncta and the total number of fluorescent cells in the FOV were counted manually.

## Structured Illumination super resolution microscopy

Super resolution images were captured using the Elyra 7 eLS lattice SIM super resolution microscope (Zeiss, Germany) with an sCMOS camera, a 63×1.46 NA water immersion objective, with a 1.6x further optical magnification. 405 nm, 488 nm and 585 nm lasers were used for excitation of the cerulean, Atto-488, and mCherry respectively. Microscope control and data acquisition was performed using ZEN Black version 3.3.89. 16-bit 2D image sets were collected with 13 phases and analyzed using the SIM^2 image processing tool by Zeiss.

## Flow cytometry measurements

IPTG / C4HSL-induced or non-induced *E. coli* BL21 cells were grown overnight, diluted 1:100 the next day and grown until OD of 0.3. Cells were diluted 1:10 in 1xPBS in a 1.5 ml tube and vortexed. Samples and appropriate controls were loaded onto a 96-wells plate (Thermo Scientific, cat. 167008) in triplicates, each well containing 100 μl of the diluted bacterial cells. The cells were then measured using flowcytometry (MACSquant VYB, Miltenyi Biotec), with the 561 nm excitation laser and the Y2 detector channel (a 615/20 nm filter). The flowcytometer was calibrated using MacsQuant calibration beads (Miltenyi Biotec) before measurement. Running buffer, washing solution, and storage solution were all purchased from the manufacturer (cat. numbers 130092747, 130092749, and 130092748, Miltenyi Biotec, respectively). Voltages for the SSC, FSC, and mCherry (Y2) channel were 467, 313, and 333 volts, respectively.

Events were defined using an FSC-height trigger of 3. An FSC-area over SSC-area gate (gate-1) was created around the densest population in the negative control (non-induced cells), and events falling inside this gate were considered live bacteria. From the gate-1-positive cells, an SSC-height over SSC-area gate (gate-2) was created along the main diagonal, and events falling inside this gate were considered single bacterium. Finally, from the gate-2-positive cells, a histogram of mCherry distribution was created, with the threshold for mCherry-positive cells set by leaving around 0.1% positive events in the negative control.

## Image analysis

The brightest spots (top 10%) in the field of view were tracked over time and space via the imageJ MosaicSuite plugin[33–35]. A typical field of view usually contained dozens of granules (in-vitro) or cells containing puncta (in vivo) (Supplementary Fig. 12a, b).

The tracking data, (x,y,t coordinates of the bright spots centroids), together with the raw microscopy images were fed to a custom built Matlab (The Mathworks, Natick, MA) script designed to normalize the relevant spot data. Normalization was carried out as follows: for each bright spot, a 14-pixel wide sub-frame was extracted from the field of view, with the spot at its center. Each pixel in the sub-frame was classified to one of three categories according to its intensity value. The brightest pixels were classified as 'spot region' and would usually appear in a cluster, corresponding to the spot itself. The dimmest

pixels were classified as 'dark background', corresponding to an empty region in the field of view. Lastly, values in between were classified as 'cell background' (Supplementary Fig. 12c). We note that for the in vitro experiments the 'dark background' and 'cell background' pixel groups yield similar intensity values. This, however, does not affect the performance of the algorithm for in vitro experiments. Classification was done automatically using Otsu's method[36]. From each sub-frame, two values were extracted, the mean of the 'spot region' pixels and the mean of the 'cell background' pixels, corresponding to spot intensity value and cell intensity value. This was repeated for each spot from each frame in the data, resulting in sequences of intensity vs. time for the spot itself and for the cell background. (Supplementary Fig. 12d).

## Signal analysis

We assume a noise model comprised of both additive and exponential components, corresponding to fluorescent proteins (bound or unbound) not relating to the spot itself, and photobleaching. This can be described as follows:

$$y(t) = (S(t) + c(t)) \cdot f(t) \tag{1.1}$$

$$c(t) = c_0(t) \cdot f(t) \tag{1.2}$$

Where $y(t)$ is the observed spot signal, $S(t)$ is the underlying spot signal which we try to extract, $c(t)$ is the observed cell background signal, $c_0(t)$ is the underlying background signal and $f(t)$ is the photobleaching component.

To find $S(t)$, we assume:

$$c_0(t) \approx c_0 = const \tag{1.3}$$

This leads to:

$$\frac{y(t)}{c(t)} = \frac{S(t) + c_0}{c_0} \tag{1.4}$$

$$S(t) = c_0\left(\frac{y(t)}{c(t)}\right) - c_0 = c_0\left(\frac{y(t)}{c(t)} - 1\right) \tag{1.5}$$

To get $y(t)$, we filter the measured spot signal with a moving average of span 13, in order to remove high frequency noise effects, and smooth out fluctuations (see section – Identifying burst events). To get $c(t)$, we fit the measured cell background signal to a 3$^{rd}$ degree polynomial (fitting to higher degree polynomials did not change the results). This is done to capture the general trend of the signal while completely eliminating fluctuations due to random noise.

## Identifying burst events

We assume the total fluorescence is comprised of three distinct signal processes: RNP granule fluorescence, background fluorescence and noise. We further assume that background fluorescence is slowly changing, as compared with granule fluorescence which depends on the dynamic and frequent insertion and shedding events occurring in the droplet. Finally, we consider noise to be a symmetric, memory-less process. Based on these assumptions, we define a "signal-burst" event as a change or shift in the level of signal intensity leading to either a higher or lower new sustainable signal intensity level. To identify such shifts in the base-line fluorescence intensity, we use a moving-average filter of 13 points (i.e., 2 min) to smooth the data. The effect of such an operation is to bias the fluctuations of the smoothed noisy signal in the immediate vicinity of the bursts towards either a gradual increase or decrease in the signal (Supplementary Fig. 13a). Random single fluctuations, which do not settle on a new baseline level are not expected to generate a gradual and continuous increase or decrease over multiple time-points in a smoothed signal. Following this, we search for contiguous segments of gradual increase or decrease and record only those whose probability for occurrence is 1 in 1000 or less given a Null hypothesis of randomly fluctuating noise.

To translate this probability to a computational threshold, we first compute the intensity difference distribution for every trace separately. This distribution is computed by collecting all the instantaneous differences in signal ($\Delta S(t_i) = S(t_i) - S(t_{i-1})$) and binning them (Supplementary Fig. 13b). Given a particular trace the likelihood for observing an instantaneous signal increase event in a time-point ($t_i$) can therefore be computed as follows:

$$P_{inc} = \frac{N(\triangle S(t_i) > 0)}{N_{tot}} \tag{1.6}$$

where $N(\Delta S(t_i) > 0)$ and $N_{tot}$ correspond to the number of increasing instantaneous events and total number of events in a trace respectively. Likewise, the number of decreasing instantaneous events is defined as:

$$P_{dec} = \frac{N(\triangle S(t_i) < 0)}{N_{tot}} \tag{1.7}$$

This in turn allows us to compute the number of consecutive instantaneous signal increase events ($m$) to satisfy our 1 in 1000 threshold for a significant signal increase burst event $m$ as follows:

$$p_{inc}^m = \frac{1}{2^{10}} \Rightarrow m \cdot \log_2(p_{inc}) = -10 \Rightarrow m = -10/\log_2(p_{inc}) \tag{1.8}$$

The threshold is calculated for each signal separately and is usually in the range of 7–13 time points. An analogous threshold is calculated for decrements in the signal and is typically in the range $[m-1, m+1]$.

To account for the presence of the occasional strong instantaneous noise fluctuations appearing in experimental signals, we allow isolated reversals in the signal directionality (e.g., an isolated one time point decrease in an otherwise continuous signal increase environment). Furthermore, since the moving average filter itself can induce correlations in the signal, we determined that the minimum allowed threshold is the moving average window span. This means that any calculated threshold lower than the moving average size is increased to this bare minimum.

We mark each trace with the number of events whose duration exceeds the threshold and define those as bursts. Segments within the signal that are not classified as either a negative or positive burst event are considered unclassified. Unclassified segments are typically signal elements whose noise profile does not allow us to make a classification into one or the other event-type. For each identified segment we record the amplitude ($\Delta I$), and duration ($\Delta t$). In Supplementary Fig. 13c we mark the classifications on a sample trace with positive "burst", negative "burst", and non-classified events in green, red, and blue, respectively. We confine our segment analysis between the first and last significant segments identified in each signal, since we cannot correctly classify signal sections that extend beyond the observed trace.

## Estimating the signal amount per slncRNA-RBP complex

Given the fact that we cannot directly infer the fluorescence intensity associated with a single RNA-RBP complex, we fitted the distributions

with a modified Poisson function of the form:

$$p(I) = \frac{\lambda^{\frac{I}{k_0}} e^{-\lambda}}{\left(\frac{I}{k_0}\right)!} \tag{1.9}$$

where $I$ is the experimental fluorescence amplitude, $\lambda$ is the Poisson parameter (rate), and $K_O$ is a fitting parameter whose value corresponds to the amplitude associated with a single RBP-bound slncRNA molecule within the burst. For each rate we chose the fit to $K_O$ that minimizes the deviation (MSE) from the experimental data. Fits were validated by observing the resulting QQ-plots.

## Statistics and reproducibility
Each granule formation reaction and subsequent microscopy experiment was successfully repeated in duplicate on multiple days as follows: PCP-3x – 4 days; PCP-4x – 3 days; PCP-8x – 3 days; PCP-3x/MCP-3x – 2 days; PCP-4x/MCP-4x – 2 days; PCP-14x/MCP-15x – 4 days.

RNase A degradation experiments were repeated in duplicate on 2 separate days. Titration experiments were repeated in triplicate on 2 separate days. Super resolution microscopy was repeated on 3 separate days. In-vivo experiments were repeated in triplicate over 5 separate days. Flow cytometry measurements were performed in triplicate.

## Reporting summary
Further information on research design is available in the Nature Portfolio Reporting Summary linked to this article.

## Data availability
All datasets used in this paper are available from: https://github.com/naorgk/slncRNA_Analysis[37]. All bacterial plasmids constructed for this work are available from the corresponding author, Roee Amit. Source data are provided with this paper.

## Code availability
All original code used in this paper, including tutorials and sample data is available from: https://github.com/naorgk/slncRNA_Analysis[37].

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

## Acknowledgements

We would like to acknowledge the Technion's LS&E staff (Tal Katz-Ezov Anastasia Diviatis, and Yael Lupu-haber) for help with sequencing and super resolution imaging experiments. In addition, the authors would like to thank professor Yoav Schectman from the Department of Bio-medical Engineering in the Technion for help with image and signal analysis. NG, NK, OW, SG, and RA received funding from the European Union's Horizon 2020 Research and Innovation Programme under grant agreement 664918 (MRG-Grammar) and 851065 (CARBP).

## Author contributions

NG designed the PP7-3x, PP7-3x/MS2-3x, PP7-4x, PP7-8x and PP7-4x/MS2-4x slncRNAs and carried out the experiments and analysis for all data. NK designed and synthesized the PP7-4x/Qβ–5x, and the PP7-14x_MS2-15x slncRNAs. OW assisted with flow cytometry experiments and data analysis, SG assisted and guided the experiments and image analysis. RA supervised the study. NG and RA wrote the manuscript.

## Competing interests

The authors declare no competing interests.
