## [Peer Review File · Nature Communications]

REVIEWER COMMENTS

Reviewer #1 (Remarks to the Author):

In this manuscript, Granik and colleagues use synthetic RNA hairpins and their binding proteins to form RNA based gel-like granules. They execute their work both in vitro and in vivo to show how the granules are formed depending on the number of hairpins contained in each structure. The hypothesis that the authors started from to drive this work is very important and of current debate in the field. The manuscript is well written and easy to read. It flows correctly and has a good potential to be published. There are several concerns that I would like to state, so that the authors could work on:

- 1- In figure 1b the authors claim that the longer slncRNA molecules exhibit larger structures. It is clear by eye that the number of the structures is higher however it is difficult to notice the size increase. Could the author do measurements of the different granule sizes in the different conditions and perform statistical analyses to prove their idea?
- 2- In the same figure, did the author use higher concentrations of PCP3x to check if it is able to form granules?
- 3- In figure 1c are the plots normalized to the amount of incorporated fluorescent U? the longer the size of the fragment, the more fluorescent uracils are incorporated and the higher the signal is.
- 4- The interpretation of the results in figure 1d and 1e should be more developed and correlated with the normalization in fig 2c.
- 5- In figure 2b, the authors should show the control RNA experiments. It is exactly the negative control that is missing for this experiment. This could help understanding/proving that the hairpins are the important structures and not any RNA with its RNA binding Protein are able to form granules
- 6- In figure 2b, could the authors use any other RNA binding protein to demonstrate that this observation is due to the specificity of PCP and not any other physical properties of RBPs.
- 7- In figure 2c and 2d the same normalization as in figure 1 should be used.
- 8- In figure 2e a titration experiment could be of great interest to understand the dynamics of the granules. Authors can use non fluorescent probe competitors at different concentration to check when the proteins are displaced.
- 9- Another competition assay can be used. Once the granules are formed, authors can incubate with different probe sizes (different labels) to check the proteins displacement etc.
- 10- In figure 3b, authors can use RNase to prove their hypothesis of the RNA degradation and the bias towards negative bursts.
- 11- In figure 3e, 3f and 3g would it be possible to perform the competition/titration experiment?
- 12- In figure 4b a negative control should be used to prove the specificity of the observation.
- 13- The images in figure 4b, 4c and 4e are very low quality.
- 14- Can the authors extend the discussion about the granule formation and the dynamics of the granules in different biological process and diseases=

Reviewer #2 (Remarks to the Author):

In the manuscript by Granik et al titled, "Formation of synthetic RNA protein granules using engineered phage-coat-protein-RNA complexes," the authors studied the role of RNA-protein interaction in the formation of phase-separated granule and its dynamics in vitro

and in bacteria. Specifically, the authors aimed to demonstrate the liquid-solid phase transitions of fluorescently tagged synthetic long noncoding RNAs (slncRNAs) with different designs of hairpin motifs by imaging, with or without addition of the corresponding cognate binding proteins for those hairpin motifs. They further claim that the degree of condensation / phase transition was dependent on parameters such as the number of hairpin motifs and concentration of the components. In addition, based on measurement from real-time live imaging, the authors suggest that the RNPs are in a constant exchange of constituents (i.e. influx or efflux of RNAs and proteins) when their components are in excess. Then, they turn to imaging bacteria to further demonstrate that when expressed in live cells, these RNPs show a similar behavior.

The authors devised a novel and original imaging-based approach to study RNA-protein interactions and their subsequent phase separation in vitro and in bacteria. The subject matter is important and the implications of their studies will advance the field. However, the work would be improved by additional controls, as elaborated below, in order to provide sufficient evidence of phase separation. While the main claims of the paper are significant, there are experimental caveats, and alternative explanations to the observed phenomena, which should be addressed in order to have confidence that the imaged particles are indeed phase separated entities.

Major points:

(1) In the RNA granule experiment presented in figure 1, in vitro transcribed RNA was resuspended in 'granule buffer' containing high salt, magnesium and PEG. Of note, under such buffer conditions and at 8.5nM concentration, RNA could form secondary structures (intramolecular) at room temperature, sufficient to create an array of RNA-RNA interactions (intermolecular) to lead to formation of RNA oligomers that will be of detectable size and interpreted as phase separated entities. It is plausible that nucleic acids such as RNA, do not 'condensate' but readily form hydrogen-bonds with stretches of complementary nucleic acids. Therefore, addition of controls such as a no-hairpin containing RNA should eliminate experimental artifacts.

(2) In addition, it is not clear whether different slncRNA designs match or differ in their fluorophore-tagged Uracil contents, which would directly affect overall fluorescence intensity. If slncRNA designs with different numbers of hairpin motifs vary in their Uracil contents, it is important to know their background levels. Images in Figure 1b suggest that the background fluorescence intensity among the different slncRNAs is variable.

(3) Without knowing a priori the number of molecules in each granule, it becomes difficult to ascertain the relationship between fluorescence intensity and the number of hairpins. Also the nonlinearity of this relationship cannot be assigned to the number of hairpins as each granule may have varying numbers of RNA molecules, confounded by longer RNA molecules oligomerizing more than short RNA.

(4) In the RNA-protein granule experiments, it is not clear how much is an RNA-mediated process or a protein-driven process- the outcome of granular formation would be affected by both components. Therefore, a negative control for each component may help quantify the relative contributions. For example, slncRNA without hairpin motifs and non-binding proteins such as mCherry may help understand the process better. Another question is whether equal molar amounts of slncRNAs have been used. As the concentration of RNA component is an important parameter, it would be helpful if the same molar amounts are used. Again, the lack of these controls makes it hard to interpret whether the imaged particles are brighter due to condensation or simple binding.

(5) In addition, some scale bars are not provided (in Figure 1b and Figure 2e), and this

makes it difficult to gauge the homogeneity or uniformity of the particles, in terms of their size and shape. For example, in Figure 2f some granules appear to be bigger than others. This may be due to a variety of factors such as those enumerated above.

(6) In cells, PP7 stem-loops and PCP have been used to visualize RNA at single molecule resolution. This is because multimerizing PP7 (24x) brings together up to 24x fluorescent PCP per RNA so that it is now bright enough for detection. When expressed in bacteria, the reason why co-expression of RNA and protein (tdPCP-mCherry) appears brighter is because the RNA behaves as a scaffold to bring PCPs within a diffraction-limited spot which now fluoresces brighter than the unbound and freely diffusing tdPCP-mCherry in the background. When the protein is expressed by itself or along with control slncRNA lacking hairpin motifs, there should not be an instance where the protein accumulates within a diffraction-limited spot inside the cell as shown in Figure 4c. The quantification of this negative control experiment would be helpful in convincing the reader.

(7) The use of measurements from real-time imaging of the fluorescent granules to study kinetics of granule formation driven by the ratio of stem loops and binding proteins and the capacitor model for the shedding is an inspired approach. The interpretation of their results and its implications in the Discussion need clarification. For instance, in line 229-230 the authors suggest that RNA degradation might contribute to the bias toward shedding of slncRNA (“...the amplitude bias towards negative events is consistent with RNA degradation and lack of transcription within the in vitro suspension, leading to a net shedding of slncRNA molecules out of the granules over time”). However, in the Discussion the authors then suggest that this system can potentially be “utilized as a programmable controlled release mechanism in suitable biological settings” (lines 422-423). The authors need to consider the impact of equilibrium and non-equilibrium models in the context of the shedding. Capacitors eventually run down unless they are in a periodically charged environment.

Author summary of changes

The reviewer focused on four main issues that had to be strengthened in the revised manuscript:

1. Using proper normalization and controls to provide an unbiased and robust description of several quantitative measurements (Reviewer #1 – comments #1-7, Reviewer #2 – comments #1-2).
2. Providing stronger evidence for the liquid-gel phase transition via additional measurements which underlies the formation of the SRNP granules (Reviewer #1 – comments #8-11, Reviewer #2 – comments #4-5).
3. Requesting additional data and analysis to more firmly establish the capacitor analogy vis-à-vis equilibrium analysis (reviewer #2 – comment #7).
4. Increase the quality of the *in vivo* data (Reviewer #1 – comments #12-13, Reviewer #2 – comments #6)

To alleviate these concerns, we carried out and provided analysis for the following additional experiments:

1. To provide a response for set of normalization and control critiques above, we did the following:
 - a. All granule intensity measurements were normalized by the number of uracils to facilitate a more unbiased comparison across *slncRNA* types. All relevant panels in Fig. 1 and 2 have been altered to display normalized data.
 - b. Negative controls including *slncRNA* lacking hairpins, non-binding fluorescent protein, and *tdMCP-mCerulean* that is incapable of binding PCP hairpins, thus strengthening the claim of specificity and relationship to number of hairpins. New SI figures have been added to show these controls
2. To provide more evidence for the liquid-gel phase transition, we did the following:
 - a. Formation of granules in the presence of RNAase. This experiment showed strong reduction in amplitude of entry events, while shedding events were maintained at the same amplitude. This established that degradation of *slncRNA* predominantly occurs in the solution and not within the granules.
 - b. Competition experiments. We added a whole new section just dedicated for protein replacement or competition experiment with *tdPCP-mCer*. Here, *tdPCP-mCer* was added at the start of the observation, and after granules with *tdPCP-mCherry* formed. These experiments demonstrate that while the two proteins rapidly equilibrate within the solution, the granules remain outside of equilibrium over the 1 hr time scale of the measurement consistent with the liquid-gel phase transitions.
 - c. Due to the volume of work involved and importance of these new results a new figure was added (replacing figure 3 in the previous manuscript)that are dedicated to the liquid-gel transition analysis (New Fig. 3 and 4).
3. The capacitor analogy was strengthened by another set of experiments where the ratio of the protein concentration to that of the *slncRNA* was systematically varied and characterized. The result here yield granules whose total amount of

stored protein depended on the protein to slncRNA concentration, which is analogous to classical capacitance. As for the liquid-gel transition, the importance of the new data necessitated adding another figure dedicated to the capacitor analog/application (New Figure 5).

- 4. Finally, Super-resolution images of the in vivo system and additional analysis are provided in new Figures 6, 7 and associated SI panels. This firmly establishes the application of increased protein titer produced within cells.*

In summary, as will be apparent from the detailed responses below, we have substantially upgraded the paper as a result of the reviewer's critiques and would like to thank them and the editor for these very helpful remarks.

REVIEWER COMMENTS

Reviewer #1 (Remarks to the Author):

In this manuscript, Granik and colleagues use synthetic RNA hairpins and their binding proteins to form RNA based gel-like granules. They execute their work both in vitro and in vivo to show how the granules are formed depending on the number of hairpins contained in each structure. The hypothesis that the authors started from to drive this work is very important and of current debate in the field. The manuscript is well written and easy to read. It flows correctly and has a good potential to be published. There are several concerns that I would like to state, so that the authors could work on:

1- In figure 1b the authors claim that the longer slncRNA molecules exhibit larger structures. It is clear by eye that the number of the structures is higher however it is difficult to notice the size increase. Could the author do measurements of the different granule sizes in the different conditions and perform statistical analyses to prove their idea?

Author response: We thank the reviewer for this suggestion. The reviewer is correct in that we have selected to display sample fields of view which contain just localized puncta, making it difficult to discern sizes by eye. We performed the statistical analysis suggested and the results are presented in the revised figure 1e. In addition, we have added a sample image showing a larger structure to better demonstrate the phenomena, presented in the revised figure 1c.

2- In the same figure, did the author use higher concentrations of PCP3x to check if it is able to form granules?

Author response: We thank the reviewer for this comment. We performed a titration experiment with increasing concentrations of PCP-3x and discovered that it does granulate on its own at higher concentrations (>20 nM). The figure pertaining to this experiment can be found in the supplementary information, and relevant discussion has been added to the manuscript. We believe that the condensation observed for PCP-3x may result from a different mechanism rather than the granulation described for the rest of the slncRNAs. This is further discussed in the response to reviewer #2 comment #1 – below.

3- In figure 1c are the plots normalized to the amount of incorporated fluorescent U? the longer the size of the fragment, the more fluorescent uracils are incorporated and the higher the signal is.

Author response: We thank the reviewer for this insight. Indeed, the amount of Uracil bases differs between the different slncRNA sequences. On par with the reviewer's suggestion, we have replotted all relevant panels (i.e. new Fig. 1d and 2d) to display a mean granule fluorescence as a normalized observable. The normalized mean fluorescence is defined as the mean granule fluorescence divided by the estimated number of fluorescent uracils in the slncRNA sequence, thus creating a standardized observable which is independent of design biases.

4- The interpretation of the results in figure 1d and 1e should be more developed and correlated with the normalization in fig 2c.

Author response: It is unclear to us what the reviewer precisely meant by this comment. The interpretation of panels 1d-e in the original manuscript (panels 1f-g in the revised manuscript) regards the slncRNA only granules. The cross-linking and quenching as a function of the number of hairpins observed for the slncRNA-only granules do not necessarily relate to the protein encasing SRNP granules.

That being said, we provide two additional pieces of analysis that may alleviate the reviewer's concern in this case. The first is presented in Fig. 2d, where we show that the quenching effect is eliminated for the SRNP granules, with the single exception of possibly the PCP-14x/MCP-15x granule. In the second, we provide an additional super-resolution microscopy image (Fig. 2e), where we show that the PCP-4x granule looks qualitatively different from the PCP-14x/MCP-15x granule (i.e. cage-like structure vs a meshed shell structure, respectively). If, nevertheless, the reviewer has additional concerns regarding this issue, we will be happy to provide an explanation.

5- In figure 2b, the authors should show the control RNA experiments. It is exactly the negative control that is missing for this experiment. This could help understanding/proving that the hairpins are the important structures and not any RNA with its RNA binding Protein are able to form granules

Author response: We thank the reviewer for this suggestion. Microscopy images depicting granule reactions with the negative control slncRNA (which does not encode for hairpin binding sites) can now be found in Figure S2 (RNA only granule reaction) and Figure S5 (RNA-protein granule reaction). Both do not show formation of granules and present as empty field of views. Consequently, granule formation depends both on the presence of >3 hairpins and on a specific hairpin binding protein.

6- In figure 2b, could the authors use any other RNA binding protein to demonstrate that this observation is due to the specificity of PCP and not any other physical properties of RBPs.

Author response: We thank the reviewer for this important suggestion. To demonstrate that granule formation is dependent on the presence of a specific hairpin-binding protein, we expressed and purified tdMCP-mCherry. Next, we carried out the granulation reaction using the new protein and PCP-8x, a class I slncRNA which does not encode for MCP binding sites, and PCP-3x/MCP-3x, class II slncRNA which does. The results show that SRNP granules only form with PCP-3X/MCP-3X, while RNA-only granules form with PCP-8x as expected. These results are discussed in the manuscript and presented in figure S6.

7- In figure 2c and 2d the same normalization as in figure 1 should be used.

Author response: We agree with the reviewer that normalizing by the number of uracil bases should be done for all relevant measurements. Consequently, we have revised panel 2d to show that normalized granule fluorescence intensity. However, panel 2c is a measurement of mCherry protein intensity, and as such we do not believe that normalizing by the number of labelled uracils is correct for this case. To provide additional clarification, we have added text to the figures specifying the measured channel (green for RNA/red for protein) to avoid any misunderstandings.

8- In figure 2e a titration experiment could be of great interest to understand the dynamics of the granules. Authors can use non fluorescent probe competitors at different concentration to check when the proteins are displaced.

Author response: We thank the reviewer for suggesting the various competition and titrations experiments in #8-#11. We agree that such experiments can provide enormous value in probing granule dynamics, and in the process firmly establishing the liquid-gel phase transition which ultimately underlies the observed dynamics. As a result, we carried out three new experiments as suggested by the reviewer:

- Observation of granule dynamics in the presence of Rnase whose results are provide in revised Figure 3c-d (discussed in #10 below)*
- Granule dynamics in the presence of a competitor provided at a higher concentration than the original tdPP7-mcherry which was used to form the granules. The data for this experiment is provided in revised Figure 4, and its implications are discussed in #9 below.*
- Protein to slncRNA ratio titration experiments. The data us provided in revised Figure 5d-f, and its implications are discussed in #11 below.*

The results of these dynamics experiments have provided another rich data set which helped to strengthen our finding of liquid-gel transition, and with it the capacitor analogy. Finally, we also added a section to the revised discussion, where we discuss the implications that emerge from these various dynamical experiments.

9- Another competition assay can be used. Once the granules are formed, authors can incubate with different probe sizes (different labels) to check the proteins displacement etc.

Author response: In order to carry out this very important experiment, we designed, expressed, and purified tdPCP-mCerulean. We then formed the granules as usual, and at the beginning of the microscopy tracing experiments added the tdPCP-mCerulean competitor at 2:1 ratio to tdPCP-mCherry. We detail the results in the revised manuscript and in revised Figure 4. In brief, the competition experiments demonstrate that the protein components rapidly equilibrate in solution. However, the granules remain not in equilibrium for the duration of the 1 hr trace, despite prominent penetration of the mCerulean competitor. The lack of equilibration in the granules provides an additional level of experimental support to the liquid-gel phase transition. We would like to thank the reviewer for this excellent suggestion, which helped to substantially improve the manuscript.

10- In figure 3b, authors can use RNase to prove their hypothesis of the RNA degradation and the bias towards negative bursts.

Author response: We would like to thank the reviewer for this suggestion as well, as it provided another independent data set which helped in further characterizing the granule gel-like dynamics. As mentioned above, the results for the Rnase experiments are provided in revised Fig. 3c-d. In this case, we opted to present the results for the slncRNA-only granules in the presence of Rnase. Here, the Rnase experiment allowed us to probe the dynamics of the denser slncRNA-only structures, which provided an independent verification of the gel-like characteristics of the slncRNA-only granules. Moreover, incubating the SRNP granules in the presence of the RNase led to rapid degradation of the granules, which did not allow us to collect a sufficiently large statistical sample. Our results show that while entry events rapidly decrease in amplitude consistent with slncRNA degradation in solution, the amplitude of the shedding events remain similar to the non-Rnase control indicating that slncRNA stored within the granules is not susceptible to degradation. This implies that either the Rnase cannot penetrate the dense slncRNA only granules, or the internal slower dynamics of the gel-like phase substantially slow its rate of catalysis. At the very least, this experiment provides important support to the claim that slncRNA-only granules also result from a liquid-gel phase transition. This finding indicates that this transition is a more general phenomenon associated with RNA decorated with multiple hairpin structures rather than one which is limited to a small subset of RNA-protein (RNP) granules.

11- In figure 3e, 3f and 3g would it be possible to perform the competition/titration experiment?

Author response: We also agreed with the reviewer's suggestion to carry out titration experiments, where the protein to slncRNA ratio is varied. The results of these experiments depicted in Fig. 5d-e provide additional support to the capacitor analogy, as an increase in the protein to slncRNA ratio leads to more frequent entry events, which in turn leads to a higher stored protein titer. This type of behavior shows that the amount of stored protein is proportional to the protein to slncRNA ratio allowing us to derive an equation similar to $Q=CV$. Here, protein concentration corresponds to the biochemical "charge Q", C is the biochemical "capacitance", and the protein to slncRNA corresponds to the biochemical "voltage". Consequently, the titration experiment strengthened the capacitor analogy allowing to discuss the implications of this analogy in additional depth in the revised discussion section. Once again, we would like to thank this reviewer for this very helpful suggestion, which helped to substantially improve the manuscript.

12- In figure 4b a negative control should be used to prove the specificity of the observation.

Author response: In the original manuscript, Figure 4 contained a sample image showing a bacterial cell which expresses the negative control slncRNA together with the tdPCP-mCherry protein, resulting in no bright spots at the cell poles. When reexamining these images, we understand the reviewer's critique as a comment on the low-resolution nature of the images, which made it difficult to tell the difference between these cells and one expression granules. In the revised figure 6, the original

images were replaced with higher resolution images obtained with a structured illumination microscopy, which clearly show a lack of granules within the bacterial cells expressing the negative control slncRNA.

13- The images in figure 4b, 4c and 4e are very low quality.

Author response: We agree with the Reviewer's critique that the images of the cells shown in the original manuscript were of poor quality. As a result, in the revised manuscript figure 6, we replaced all of these images with taken using a structured illumination super resolution microscope.

14- Can the authors extend the discussion about the granule formation and the dynamics of the granules in different biological process and diseases=

Author response: The question of relevance to natural biological processes is always a fraught and controversial issue when synthetic biology experiments such as the one presented in this manuscript are involved. While it is obvious that our synthetic system has little relevancy to naturally occurring granules, speckles, or p-bodies, the recent paper by Jain and Vale (<https://doi.org/10.1038/nature22386>) discusses the potential role of RNA in repeat expansion disorders. This discussion is consistent with the type of liquid-gel transition observed here on a similarly structured RNA molecule, and thus may be related. Consequently, we feel comfortable making a passing mention in the discussion as to the potential relevancy of our findings to repeat expansion disorders. Regarding naturally occurring biological processes, since most biologically relevant phase separated bodies are thought to be liquid like, our investigation into gel type phase separation remains a different, but parallel topic which may or may not be relevant to many of these naturally occurring systems. Whether a broader liquid-gel transition is associated with naturally occurring RNA-containing bodies such as paraspeckles remains to be seen and is a question beyond the scope of this paper.

Reviewer #2 (Remarks to the Author):

In the manuscript by Granik et al titled, "Formation of synthetic RNA protein granules using engineered phage-coat-protein-RNA complexes," the authors studied the role of RNA-protein interaction in the formation of phase-separated granule and its dynamics in vitro and in bacteria. Specifically, the authors aimed to demonstrate the liquid-solid phase transitions of fluorescently tagged synthetic long noncoding RNAs (slncRNAs) with different designs of hairpin motifs by imaging, with or without addition of the corresponding cognate binding proteins for those hairpin motifs. They further claim that the degree of condensation / phase transition was dependent on parameters such as the number of hairpin motifs and concentration of the components. In addition, based on measurement from real-time live imaging, the authors suggest that the RNPs are in a constant exchange of constituents (i.e. influx or efflux of RNAs and proteins) when their components are in excess. Then, they turn to imaging bacteria to further demonstrate that when expressed in live cells, these RNPs show a similar behavior.

The authors devised a novel and original imaging-based approach to study RNA-protein interactions and their subsequent phase separation in vitro and in bacteria. The subject matter is important and the implications of their studies are will advance the field. However, the work would be improved by additional controls, as elaborated below, in order to provide sufficient evidence of phase separation. While the main claims of the paper are significant, there are experimental caveats, and alternative explanations to the observed phenomena, which should be addressed in order to have confidence that the imaged particles are indeed phase separated entities.

Major points:

(1) In the RNA granule experiment presented in figure 1, in vitro transcribed RNA was resuspended in 'granule buffer' containing high salt, magnesium and PEG. Of note, under such buffer conditions and at 8.5nM concentration, RNA could form secondary structures (intramolecular) at room temperature, sufficient to create an array of RNA-RNA interactions (intermolecular) to lead to formation of RNA oligomers that will be of detectable size and interpreted as phase separated entities. It is plausible that nucleic acids such as RNA, do not 'condensate' but readily form hydrogen-bonds with stretches of complementary nucleic acids. Therefore, addition of controls such as a no-hairpin containing RNA should eliminate experimental artifacts.

Author response: We agree with the reviewer's concerns and appreciate the constructive comments. The reviewer is correct in that a multitude of forces can drive the condensation of the RNA molecules. To demonstrate the dependence of condensation on the presence of hairpin structures, we have added a no hairpin slncRNA negative control containing no designed hairpins but with a GC content similar to the PCP-14x/MCP-15x slncRNA. Sample field of view images are shown in Figure S2 and S5. The images show that no granules form both in the RNA-only and RNA-protein granule reaction conditions. In addition, we wish to refer the reviewer to figure S1, showing a titration experiment with the PCP-3x slncRNA where RNA condensates form only at higher concentrations (20 nM and above), where with other slncRNAs granules were observed at 8.5 nM. Consequently, condensation of our slncRNA is dependent on the presence of at least 3 hairpin structures.

(2) In addition, it is not clear whether different slncRNA designs match or differ in their fluorophore-tagged Uracil contents, which would directly affect overall fluorescence intensity. If slncRNA designs with different numbers of hairpin motifs vary in their Uracil contents, it is important to know their background levels. Images in Figure 1b suggest that the background fluorescence intensity among the different slncRNAs is variable.

Author response: We thank the reviewer for this comment and helpful suggestion. Indeed, the different slncRNAs differ in their uracil contents. To eliminate this technical artefact, we followed the suggestion of reviewer #1 (see response #3 above and new figures 1c and 2d), and have replaced all the relevant results to data normalized by the putative number of labeled uracil bases in each slncRNA molecule (assuming a 35% labeling efficiency per the manufacturer's protocol). In addition, we also measured the background fluorescence per the reviewer's request and the results are presented in figure S3a. The data shows that while the background is indeed somewhat variable between different slncRNA types, all background levels cluster between a narrow range of fluorescence values (350 to 450 AU). This variability (i.e. ± 50 AU) is at least an order of magnitude smaller than the signal of the granules, and thus cannot affect the interpretation of the results. Finally, we wish to further note that our dynamic tracking algorithm accounts for local background levels. Therefore, the amplitude data in figures 3 and 4 should be relatively unaffected by global variability in background fluorescence.

(3) Without knowing a priori the number of molecules in each granule, it becomes difficult to ascertain the relationship between fluorescence intensity and the number of hairpins. Also the nonlinearity of this relationship cannot be assigned to the number of -hairpins as each granule may have varying numbers of RNA molecules, confounded by longer RNA molecules oligomerizing more than short RNA.

Author response: In general, we agree with the reviewer that without having a good estimate for the number of molecules within the granules it is hard to estimate the effect of other parameters (e.g. hairpin on expression). This is the reason which ultimately led to the usage of labelled uracils in this experiment. We would like to point the reviewer's attention to figure 5b in the revised manuscript, which provides an estimate to the distribution of the number of slncRNAs within each granule. This estimate assume that the mean burst amplitude measured across a 1hr trace per granule corresponds to the fluorescence of a single molecule of slncRNA. This plot shows that for type II granules, the median number of slncRNAs is approximately the same for all granules. Hence, oligomerization of longer slncRNA molecules does not seem occur, and the most reasonable explanation for the variation observed in figure 5a for the same set of molecules is a dependence on the number of hairpins.

(4) In the RNA-protein granule experiments, it is not clear how much is an RNA-mediated process or a protein-driven process- the outcome of granular formation would be affected by both components. Therefore, a negative control for each component may help quantify the relative contributions. For example, slncRNA without hairpin motifs and non-binding proteins such mCherry may help understand the process better. Another question is whether equal molar amounts of slncRNAs have been used. As the concentration of RNA component is an important parameter, it would be helpful if the same molar amounts are used. Again, the lack of these controls

makes it hard to interpret whether the imaged particles are brighter due to condensation or simple binding.

Author response: We agree with the reviewer that these controls are necessary to establish the primacy of the slncRNA in granule formation. Consequently, we have added the requested negative controls to the supplementary material, as well as relevant discussion in the main text. Figure S2 and S5 present sample microscopy fields of view from an RNA-only and RNA-protein granule reactions, prepared with the negative control slncRNA which does not encode for any hairpin structures to the best of our knowledge. In addition, we expressed and purified tdMCP-mCherry protein and formed granules with PCP-8x, a class I slncRNA which does not encode for MS2 binding sites, and PCP-3x/MCP-3x, a class II slncRNA which does. The results show that tdMCP-mCherry does not bind PCP-8x, while it does bind PCP-3x/MCP-3x as expected. Consequently, this experiment shows that SRNP granules form only if the specific hairpin-binding protein is present. Due to the importance of this control, these results are discussed in the revised manuscript and presented in figure S6.

With regards to the equal Molar amount question, it is unclear to us what precisely the reviewer meant. If the meaning was to carry out a granulation reaction with equ-Molar amount of slncRNA and protein, or equ-Molar amount of hairpins and protein, this data now is now provided in new Figure 5d-f (and discussion therein). If, however, the reviewer meant that slncRNAs in different experiments will be tested in equal Molar amounts, we used the following conditions:

- All slncRNA data in Figure 1 was taken with an initial slncRNA concentration of 8.5 nM in the granulation reaction.*
- For the SRNP granule experiments slncRNA concentrations used were in the 10-20 nM range, as is now written in the text.*

(5) In addition, some scale bars are not provided (in Figure 1b and Figure 2e), and this makes it difficult to gauge the homogeneity or uniformity of the particles, in terms of their size and shape. For example, in Figure 2f some granules appear to be bigger than others. This may be due to a variety of factors such as those enumerated above.

Author response: We thank the reviewer for this important comment. We have added scale bars to all fluorescence microscopy images. Unless stated otherwise, all scale bars are 10 μ m.

(6) In cells, PP7 stem-loops and PCP have been used to visualize RNA at single molecule resolution. This is because multimerizing PP7 (24x) brings together up to 24x fluorescent PCP per RNA so that it is now bright enough for detection. When expressed in bacteria, the reason why co-expression of RNA and protein (tdPCP-mCherry) appears brighter is because the RNA behaves as a scaffold to bring PCPs within a diffraction-limited spot which now fluoresces brighter than the unbound and freely diffusing tdPCP-mCherry in the background. When the protein is expressed by itself or along with control slncRNA lacking hairpin motifs, there should not be an instance where the protein accumulates within a diffraction-limited spot inside the cell as shown in Figure 4c. The quantification of this negative control experiment would be helpful in convincing the reader.

Author response: We thank the reviewer for this note. In the original manuscript, figure 4 contained a sample image showing a bacterial cell which expresses the negative

control slncRNA together with the tdPP7-mCherry protein, resulting in no bright spots at the cell poles. As was pointed out by reviewer 1, the images were of low quality. Therefore, in the revised figure 6, the images were replaced with structured illumination microscopy images, showing this more succinctly. The image depicting bacteria expressing the negative control RNA (figure 6c) together with the tdPCP-mCherry shows uniform fluorescence in the cells. In contrast, the bacteria expressing PCP-24x slncRNA together with tdPCP-mCherry (figure 6b) shows regions of higher fluorescence in the cell poles consistent with granule formation and accumulation of proteins within granules.

(7) The use of measurements from real-time imaging of the fluorescent granules to study kinetics of granule formation driven by the ratio of stem loops and binding proteins and the capacitor model for the shedding is an inspired approach. The interpretation of their results and its implications in the Discussion need clarification. For instance, in line 229-230 the authors suggest that RNA degradation might contribute to the bias toward shedding of slncRNA (“...the amplitude bias towards negative events is consistent with RNA degradation and lack of transcription within the in vitro suspension, leading to a net shedding of slncRNA molecules out of the granules over time”). However, in the Discussion the authors then suggest that this system can potentially be “utilized as a programmable controlled release mechanism in suitable biological settings” (lines 422-423). The authors need to consider the impact of equilibrium and non-equilibrium models in the context of the shedding. Capacitors eventually run down unless they are in a periodically charged environment.

Author response: We especially thank the reviewer for this very thoughtful comment, which have led us to consider how removing our system from equilibrium can lead to additional characterization of the liquid and gel phases. In particular, both the RNase and the competition experiments probe this question. In both cases, the introduction of the RNase and protein competitor lead to rapid equilibration in the liquid phase. However, the granule phase remains outside of equilibrium for the duration of the experiment. For the competitor experiment, the amount of competitor within the granule remains below the expected value if the granule was able to achieve equilibrium. For the RNase experiment, the entry amplitudes rapidly decline in the first 30' indicating slncRNA degradation within the liquid phase, while the shedding amplitudes do not decline markedly supporting a much slower degradation process within the granules.

In general, the granule phase can be considered as either a dense liquid, gel, glass, or solid phase. The dense liquid phase should equilibrate nearly as rapidly as the dilute liquid phase, which is not the case. The glass and solid phases should not be affected at all by the introduction of the RNase and competitor as such phases do not exchange material with the surrounding fluid phase. The slow equilibration process occurring for the granule is, therefore, most consistent with a gel-like phase. Consequently, these experiments and using equilibrium considerations have substantially aided our conclusion that that what is observed in our experiments is a liquid-gel phase transition.

As a result of the importance of this subject, we devote a significant amount of the revised results and discussion sections to highlighting these equilibrium considerations, and their importance for our thesis.

REVIEWERS' COMMENTS

Reviewer #1 (Remarks to the Author):

The authors have answered all the concerns that were raised and gave a modified version of the manuscript. This version is ready to be submitted.

Reviewer #2 (Remarks to the Author):

In the revised version of the manuscript "Formation of synthetic RNA protein granules using engineered phage-coat-protein-RNA complexes" by Granik et al., it is clear that the authors made an effort to address the reviewers' comments and to make appropriate modifications to answer some concerns that were raised and provide clarifications. We find that the current version of the manuscript is ready for publication. However, we do have a few minor comments:

1. The contrast in the PCP-3x panel in Fig. 1b is so high that it looks manipulated. Please show a proper image of the background fluorescence in dynamic range.
2. Lines 108-109, it should say "(Figure 1d)".
3. Figure 2b - consider adding images of the channels separately (potentially can go to SUP figure) - it is very hard to judge the results only based on the merged images.
4. In lines 149-151 the authors write: "Unexpectedly, PP7-3x granules were witnessed in the presence of the protein, implying that tdPCP-mCherry adds a measure of multivalency to the system, and thus triggers condensation of RNA molecules that do not phase separate on their own at the concentration used".
Although not highly likely, an alternative explanation is still plausible; that the presence of 3 adjacent PP7 stem loop motives can now co-localize enough mCherry proteins in order to create a bright puncta (bringing it above background level). Bringing forward such an alternative explanation can reduce some confusion among the readers, without hurting the author's conclusions from this experiment. In addition, including the separate channel images as suggested in point 3 can help clarify some of this.
5. Line 163 - the authors write "The mCherry protein" - I would add that it is the PCP and not the MCP, as the sentence before discussed the MCP and this can confuse the reader.
6. The authors use fluorescence quenching (lines 112, 136, 177 and 180) to justify several observations throughout the manuscript. It would be more compelling if the authors could calculate empirically the estimated measure of the observed quenching from the granules.
7. Line 174 - please add a reference to the relevant figure panel.
8. The explanation for Fig 3a is unclear (lines 225-228). The observation of steady-state exchange of SRNP should not be interpreted as a 'burst.' The mean fluorescence amplitude in Fig. 3d typically hover around 1 or less and ultimately trend downward over time. This appears simply as exchange of SRNP into and out of the granule.

Reviewer #1 (Remarks to the Author):

The authors have answered all the concerns that were raised and gave a modified version of the manuscript. This version is ready to be submitted.

Reviewer #2 (Remarks to the Author):

In the revised version of the manuscript “Formation of synthetic RNA protein granules using engineered phage-coat-protein-RNA complexes” by Granik et al., it is clear that the authors made an effort to address the reviewers’ comments and to make appropriate modifications to answer some concerns that were raised and provide clarifications. We find that the current version of the manuscript is ready for publication. However, we do have a few minor comments:

1. The contrast in the PCP-3x panel in Fig. 1b is so high that it looks manipulated. Please show a proper image of the background fluorescence in dynamic range.

Author response: The image was corrected.

2. Lines 108-109, it should say “(Figure 1d)”.

Author response: The error was corrected.

3. Figure 2b - consider adding images of the channels separately (potentially can go to SUP figure) - it is very hard to judge the results only based on the merged images.

Author response: We thank the reviewer for this suggestion. The images now appear in figure S5. The relevant references in the manuscript have been updated.

4. In lines 149-151 the authors write: “Unexpectedly, PP7-3x granules were witnessed in the presence of the protein, implying that tdPCP-mCherry adds a measure of multivalency to the system, and thus triggers condensation of RNA molecules that do not phase separate on their own at the concentration used”. Although not highly likely, an alternative explanation is still plausible; that the presence of 3 adjacent PP7 stem loop motives can now co-localize enough mCherry proteins in order to create a bright puncta (bringing it above background level). Bringing forward such an alternative explanation can reduce some confusion among the readers, without hurting the author’s conclusions from this experiment. In addition, including the separate channel images as suggested in point 3 can help clarify some of this.

Author response: We thank the reviewer for this comment. We have added the proposed explanation to the manuscript.

5. Line 163 – the authors write "The mCherry protein" - I would add that it is the PCP

and not the MCP, as the sentence before discussed the MCP and this can confuse the reader.

Author response: We have revised the text as requested to provide the necessary clarification.

6. The authors use fluorescence quenching (lines 112, 136, 177 and 180) to justify several observations throughout the manuscript. It would be more compelling if the authors could calculate empirically the estimated measure of the observed quenching from the granules.

Author response: We thank the reviewer for this suggestion. Using the data from figure 2d we have estimated the quenching effect as a ratio between the expected fluorescence and the observed fluorescence, providing a numerical value for the decline in fluorescence per siRNA. Details have been added to the caption in figure 2d, the methods section, as well as to all the positions in the text where quenching is referenced as listed by the reviewer.

7. Line 174 – please add a reference to the relevant figure panel.

Author response: A reference was added to figure 2c.

8. The explanation for Fig 3a is unclear (lines 225-228). The observation of steady-state exchange of SRNP should not be interpreted as a 'burst.' The mean fluorescence amplitude in Fig. 3d typically hover around 1 or less and ultimately trend downward over time. This appears simply as exchange of SRNP into and out of the granule.

Author response: The reviewer is correct that when considering the data presented in figure 3d, the term "burst" might not be correct for the siRNA measurements. However, in the interest of establishing a nomenclature fitting to both siRNA and protein measurements, we opted to keep the term to describe movement of molecules into or out of the granule.

Considering the reviewer's comment, we have added a clarification in the text specifically stating that "signal bursts" are our own terminology.